# Adaptive spatial-temporal information processing based on in-memory attention-inspired devices

Jiong Pan [1,2,6], Fan Wu[1,2,3,6], Kangan Qian[4,5,6], Kun Jiang[4,5], Yanming Liu[1,2], Zeda Wang[1,2], Pengwen Guo [1,2], Jiaju Yin[1,2], Diange Yang[4,5] ✉, He Tian [1,2] ✉, Yi Yang[1,2] ✉ & Tian-Ling Ren [1,2] ✉

Spatial-temporal information perception is widely used for motion processing in dynamic scenes, but present technology requires relatively huge hardware resource consumption. The attention mechanism helps the human brain extract required information from tremendous data at a low cost. Here, we propose an attention-inspired artificial intelligence architecture based on hetero-dimensional modulations between zero-dimensional contact and two-dimensional electrostatic interfaces. An adaptive spatial-temporal information processing primitive is successfully implemented based on in-memory analog computing. Experiments of attention adjustments responding to different situations validate the adaptation capability to environmental changes. A demonstration of 5×5-unit data stream processing is conducted, and intensities of spatial and temporal information are varied with attention distribution from 0% to 100%. The attention-inspired device is applied to autonomous driving edge intelligence scenarios, showing high adaptability to traffic scene variations. The proposed architecture exhibits a tens-fold latency reduction, hundreds-fold area improvement, and thousands-fold energy saving compared to the conventional transistor-based circuit.

The wave of edge intelligence leads to more requirements for highly efficient spatial-temporal information perception hardware[1]. Conventional hardware solutions require complex pathways and separated equipment for sequential data storage, transmission, and processing, which lead to large time latency and energy costs[2,3]. In contrast with conventional electronics, the human brain understands spatial and temporal information from the surroundings at an extremely low cost[4,5]. The attention mechanism is used to extract significant information from tremendous data, and attention is dynamically adjustable with varying situations to ensure persistently effective information extraction in ever-changing environments[6,7]. In the human brain,

frontoparietal attention networks adjust the attention in response to present situations, and direct regions in sensory cortexes to focus on specific types of information[8–11]. The attention mechanism achieves complete information perception and significant information enhancement.

Artificial intelligence hardware designs that mimic cognition approaches of the brain have emerged in recent years based on two-dimensional (2D) materials, exhibiting high operation speed and low power consumption[12–15]. 2D neuromorphic devices and systems have been developed for in-memory analog multiplication used for artificial neural networks[16–19], and emulations of the brain's neuron and synapse

[1]School of Integrated Circuits, Tsinghua University, Beijing, China. [2]Beijing National Research Center for Information Science and Technology (BNRist), Tsinghua University, Beijing, China. [3]Shanghai Frontiers Science Research Base of Intelligent Optoelectronics and Perception, Institute of Optoelectronics, College of Future Information Technology, Fudan University, Shanghai, China. [4]School of Vehicle and Mobility, Tsinghua University, Beijing, China. [5]State Key Laboratory of Intelligent Green Vehicle and Mobility, Tsinghua University, Beijing, China. [6]These authors contributed equally: Jiong Pan, Fan Wu, Kangan Qian. ✉e-mail: ydg@tsinghua.edu.cn; tianhe88@tsinghua.edu.cn; yiyang@tsinghua.edu.cn; RenTL@tsinghua.edu.cn

functions to perform artificial synaptic information processing[20–23]. Temporal information perception hardware has been investigated based on 2D neuromorphic devices. Temporal summation hardware performs an analog weighted summation of spatial mappings at individual time into the last frame based on fading memory characteristics of devices[24–27]. The strategy is much effective for single object detection with a dark background, but object confusion and information cover-up problems exist in luminous scenes. Temporal difference hardware extracts moving objects from the background by frame-wise subtraction[28–31]. However, spatial information is lost after temporal information extraction, and the recognition capacity is limited. Complete information perception is hard to be in situ achieved based on present neuromorphic mechanisms, and a novel physical mechanism for artificial intelligence hardware to emulate the brain's attention mechanism should be developed.

This work proposes an attention-inspired device for in situ spatial-temporal information processing based on hetero-dimensional modulations. Zero-dimensional (0D) interface exhibits non-volatile state transfer behavior for data storage, and adjustable weighted analog computing is performed between input and stored data based on intrinsic interactions of 0D-2D hetero-dimensional interfaces. The attention-inspired device delivers a large state transfer ratio ($10^9$) and a high on/off shunt current range ($10^8$). An adaptive spatial-temporal information processing primitive is implemented based on reconfigurable properties of the attention-inspired device to perform attention distribution and determination computing functionalities. The attention is dynamically adjusted by situation variations. Adaptive information processing for a data stream with 5×5 units is conducted. Based on the validations, we demonstrate the attention-inspired device used for highly adaptive edge equipment. Attention-enhanced equipment exhibits full range adjustable spatial and temporal attention. A 190-fold reduction of area, 47-fold reduction of time latency, and 1411-fold reduction of energy consumption are achieved by the attention-inspired device in comparison with the conventional transistor.

## Results

### Attention-inspired information perception architecture

Here, attention-inspired devices are fabricated to emulate the brain's attention mechanism (Fig. 1a). 2D monolayer $MoS_2$ is grown by chemical vapor deposition (CVD) and is fabricated as the channel. $Ag^+$ ions with large conductivity and diffusivity are suitable for the top electrode to form the filament[32], and 0D contact interfaces between 2D transition metal dichalcogenides and Ag filament exhibit excellent contact properties and non-volatile programmable capabilities[33,34]. The scanning electron microscope image of a fabricated attention-inspired device, and corresponding transmission electron microscopy and energy dispersive spectroscopy characterizations are shown in Fig. 1b.

The attention-inspired device implements in-memory analog spatial-temporal computing based on interactive modulations of the 0D-2D hetero-dimensional interfaces. An attention-inspired adaptive spatial-temporal information processing architecture is illustrated in Fig. 1c. The attention-inspired device network for determination computing emulates frontoparietal attention networks to dynamically adjust and output the optimized attention. The attention distribution network mimics attention-controlled information enhancement of sensory cortexes, receives the information from a data stream, and outputs a single frame of data containing both spatial and temporal information. The intensities of spatial and temporal information in the output are adjusted by the attention distribution. In practical cases, both spatial and temporal information are required, and one type of information should be intensified to enhance recognition performance in specific situations. When spatial attention is increased, spatial information, including the motionless traffic light and the red body of

the bus, is enhanced. Conversely, when temporal attention is enlarged, temporal information, including the outline of the moving bus, is enhanced.

Figurs 1d, e illustrate fundamental mechanisms of the attention-inspired device. For determination network, the attention-inspired device network exhibits a neuromorphic behavior (Fig. 1d). Weight data is stored by filament states of each device unit. Situations are encoded as logic signals and input to the network. Situation descriptions include: the vehicle has a high speed, there is heavy traffic on the road, etc. True (logic 1) or False (logic 0) indicates whether the situation description is real. The unit is resting when the situation is false, and is excitatory (w/-filament state) or inhibitory (w/o-filament state) when the situation is true. Network computing of multiple units is implemented to obtain optimized attention. For the attention distribution network, input data is sequentially stored in the attention-inspired device array by filament states (Fig. 1e). The present input data ($t_2$) interacts with the previously stored data ($t_1$). The transport curve of w/-filament state (red lines) is modulated by attention to adjust output spatial and temporal information intensities.

### Attention-inspired device principles

Figure 2 illustrates the working mechanisms of the attention-inspired device. The Ag electrode (IN) is connected to the input data stream, and the control gate (CG) continuously adjusts the 0D-2D hetero-dimensional modulation characteristics (Fig. 2a). In writing mode, the input voltage $V_{IN}$ is applied to IN, and the drain is grounded (Fig. 2b). The CG voltage $V_{CG}$ controls the on-state current by buried gate electrostatic modulation of the $MoS_2$ channel. 0D contact interface between $MoS_2$ and Ag filament is formed by applying positive input voltage $V_{IN}$, and is ruptured by negative $V_{IN}$. Output characteristics curves of the $MoS_2$ channel are illustrated in Fig. 2c. The $MoS_2$ channel transfer curves are shown in Supplementary Fig. 1. Programmable filament state transfer curve of the attention-inspired device is illustrated in Fig. 2d. $V_{IN}$ is continuously scanned from path 1 to 4. Filament is formed when scanning $V_{IN}$ from 0.00 to 3.50 V, and is ruptured for $V_{IN}$ from 0.00 to −1.50 V. Compared to two-terminal filament formations[35], filament forming by a semiconductor channel can realize lower activation time and parasitic capacitances, whereby promotes the stability of filament state transfer processes[36]. The $MoS_2/HfO_2/Ag$ structure implements semiconductor-controlled filament state transfer without peripheral transistor circuits, eliminating interconnect circuits between transistors and memory devices. Filament states are stable and remain unchanged within an appropriate large $V_{IN}$ interval (−1.20 V to 2.00 V) and after removing the voltage supply. A high state transfer ratio ($10^9$) of the drain current $I_D$ that is largely higher than the semiconductor channel hysteresis (Supplementary Fig. 2) enables stable electrostatic modulation in w/o-filament state, and moderate shunt current induced by the 0D interface in w/-filament state. The shunt current $I_{IN}$ in w/-filament state is modulated by $V_{CG}$, and enables the attention-inspired device to have bilateral bound characteristics in w/-filament state (Fig. 2e). The bilateral bound behavior is necessary for attention distribution computing. When $V_{CG} < -3.00$ V, $I_{IN} < 10$ nA and the channel is cut off. For $V_{CG}$ from −3.00 to 3.00 V, $I_{IN}$ is increased with $V_{CG}$. When $V_{CG} > 3.00$ V, $I_{IN}$ is limited by the 0D interface resistance, and the variation of $I_{IN}$ with $V_{CG}$ is disabled. Experiments of filament state transfer processes and device performance have been implemented to verify stable and repeatable data storage functionalities (Supplementary Note 1). The device stability is relevant to the degradation effects of the dielectric film during the repeated filament-forming and rupture processes. Strategies to mitigate the degradation and enlarge the endurance have been reported[37–39]. The performance of device would also be affected by external factors including proton or moisture that can be incorporated into dielectric films[40,41]. More discussions on device stability are provided in Supplementary Note 2.

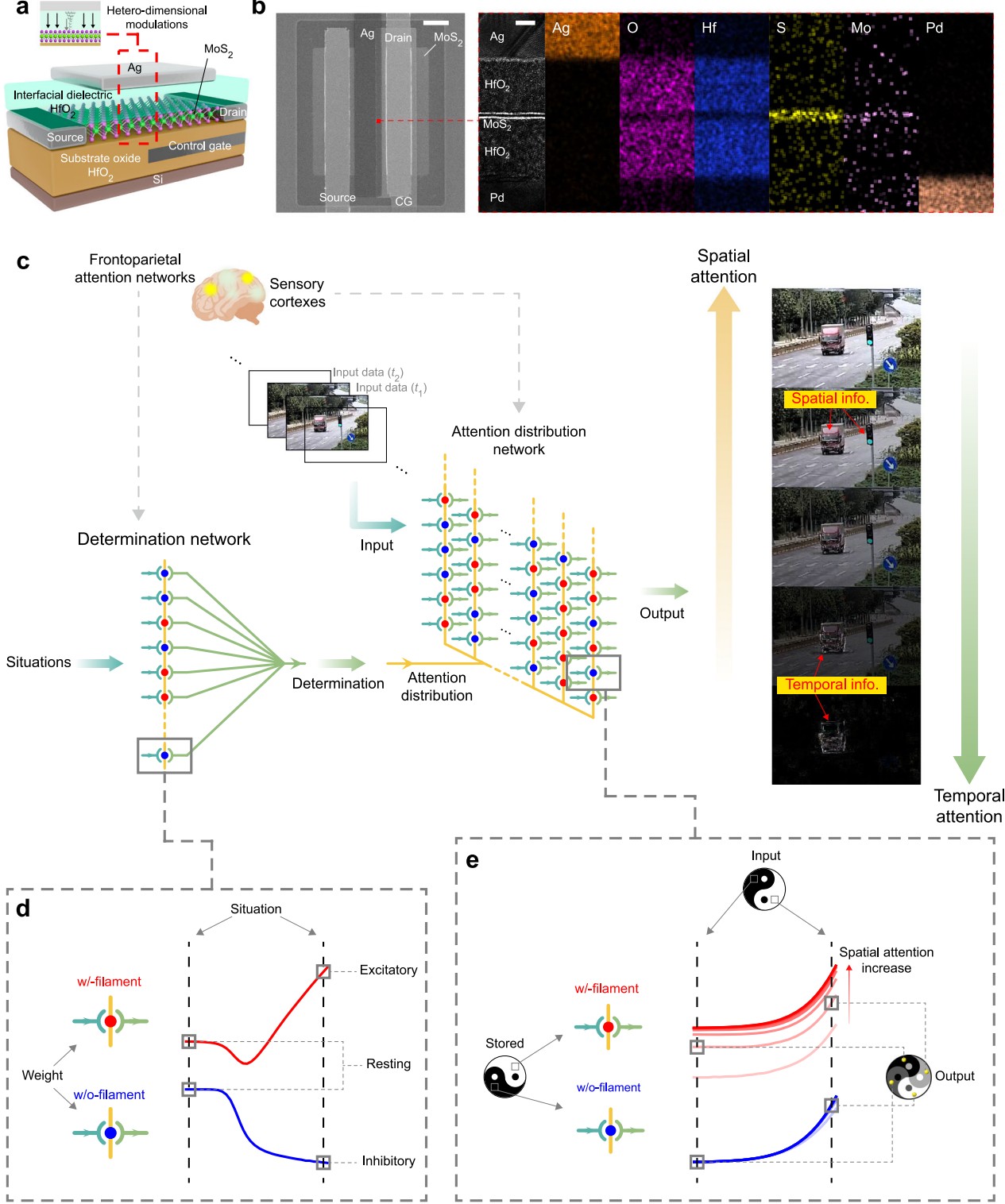

**Fig. 1 | Adaptive spatial-temporal information processing architecture based on attention-inspired devices. a** Schematic of an attention-inspired device. The highlighted structure implements 0D-2D hetero-dimensional modulations. The gray and black arrows represent 0D and 2D interfacial modulations. **b** Scanning electron microscopy characteristics of a fabricated attention-inspired device (Scale bar, 10 μm) and the corresponding transmission electron microscopy and energy dispersive spectroscopy mapping (Scale bar, 5 nm). **c** Schematic of the adaptive spatial-temporal information processing architecture. Determination network and attention distribution network perform functionalities of frontoparietal attention networks and attention-controlled sensory cortexes respectively. The output is a single-frame matrix containing adjustable spatial and temporal information. **d** An attention-inspired device unit in determination network with bidirectional responses to the situation. **e** An attention-inspired device unit in attention distribution network. In-memory adjustable analog computing of stored and input data is in situ conducted.

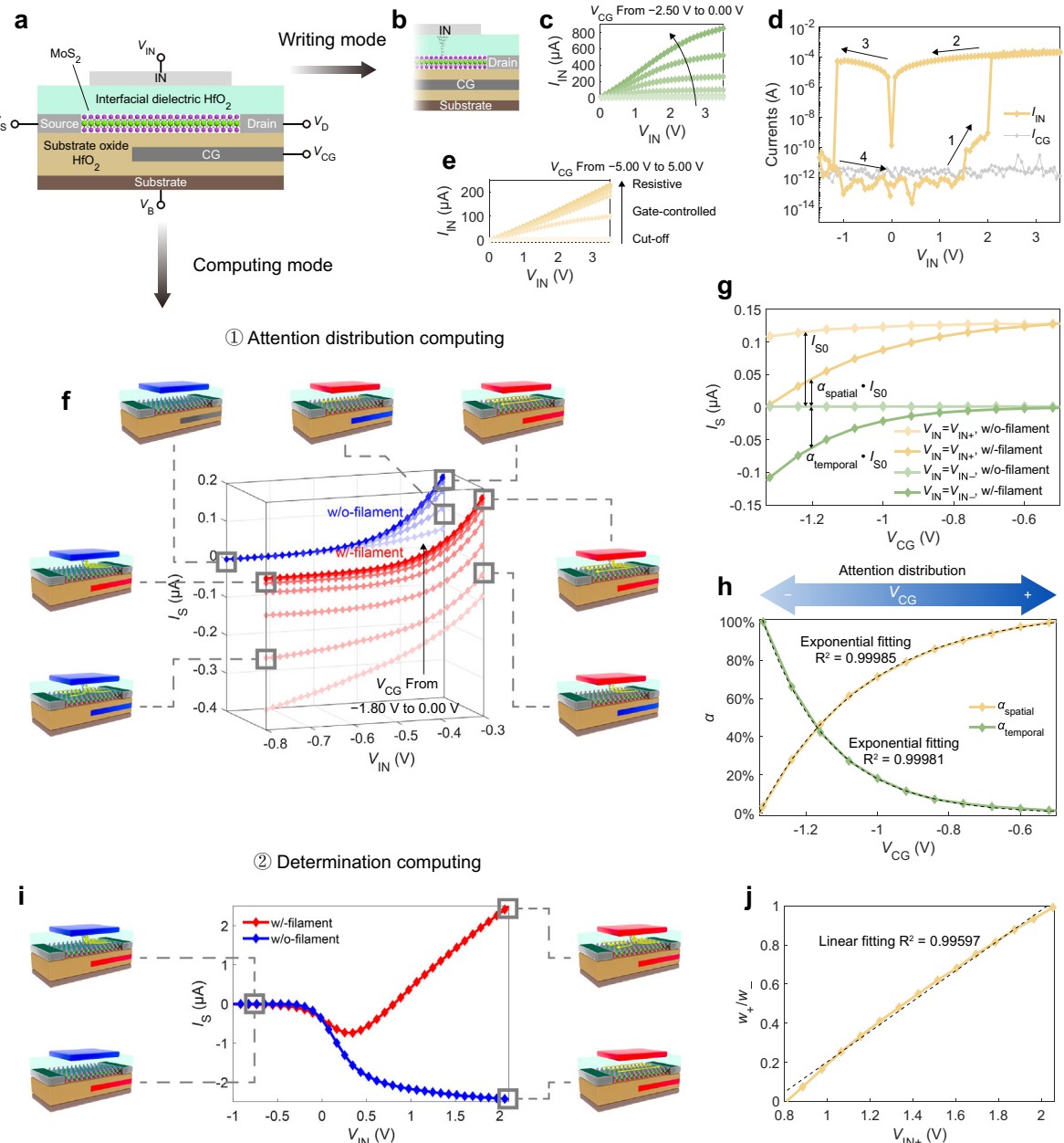

**Fig. 2 | Working mechanisms of the attention-inspired device. a** Structure of an attention-inspired device working in writing and computing modes. Writing mode: **b** Writing mode configuration. **c** Output characteristics of the MoS$_2$ channel. The drain-to-source voltage $V_{DS} = 1.00$ V. **d** Attention-inspired device filament state transfer curve. The voltage scanning path is from 1 to 4. The control gate voltage $V_{CG} = 5.00$ V. **e** Filament shunt current modulation characteristics with cut-off, gate-controlled, and resistive regions varying with $V_{CG}$. Computing mode (attention distribution computing): **f**, $I_S - V_{IN}$ transport curves under different $V_{CG}$ from −1.80 to 0.00 V. $I_S$ is the source current. Device structure schematics illustrate states of the attention-inspired device and current directions under each voltage

configuration. IN and CG electrode voltage configurations are represented by blue (low voltage), red (high voltage), or gray (arbitrary voltage) colors. **g**, $I_S - V_{CG}$ attention distribution characteristics. The input voltages $V_{IN-} = -0.72$ V, $V_{IN+} = -0.36$ V. **h**, Spatial and temporal attention varying with $V_{CG}$ that exhibit dynamic attention adjustment properties. Dash lines show the exponential fitting. The source voltage $V_S = -0.20$ V in (**f**−**h**). Computing mode (determination computing): **i** Bidirectional determination computing curves of the attention-inspired device. **j** Weight plasticity adjusted by $V_{IN+}$. The dash line shows the linear fitting. $V_S = 0.20$ V and $V_{CG} = 0.60$ V in **i** − **j**. $V_D = 0.00$ V and the substrate voltage $V_B = -1.00$ V in (**f**−**j**).

In computing mode, the attention-inspired device exhibits reconfigurable properties to implement attention distribution and determination computing modes by the source voltage $V_S$ and the CG voltage $V_{CG}$. $V_{CG}$ is tunable for attention adjustment. For attention distribution computing, transport curves varying with $V_{CG}$ in different filament states to emulate attention-directed perception in the brain are illustrated in Fig. 2f (positive direction of the source current $I_S$ is

from drain to source). In w/o-filament state (blue lines), current flows from drain to source, and bidirectional gating of $V_{IN}$ and $V_{CG}$ co-modulates $I_S$. $V_{IN}$ controls the electron injection from source to channel, thereby determining the upper bound of $I_S$ (dark blue lines in Fig. 2f). $V_{CG}$ forms and modulates the potential barrier between the CG- and IN-controlled homojunction. When $V_{CG} < V_{IN}$, the large homojunction barrier is dominant to limit electron carrier density in the

channel, so $I_S$ is increased with $V_{CG}$. When $V_{CG} > V_{IN}$, the homojunction barrier is forwardly biased. With fixed $V_{IN}$, the electron carrier density is unchanged by $V_{CG}$, and $I_S$ is saturated. The saturation characteristics in w/o-filament state provide an interval of stable $I_S$. In w/-filament state (red lines), shunt currents induced by 0D interface modulate $I_S$ transport behavior. $V_{CG}$ controls the potential of 0D interface ($\phi_{0D}$) by modulating the channel between drain and 0D interface, and $V_{IN}$ controls the 0D interface shunt current that is determined by the difference between $\phi_{0D}$ and $V_{IN}$ based on Ohms Law. When $V_{CG} = -1.80$ V, the CG-controlled channel is bent upward to reduce the flow of electron carrier from 0D interface to drain. As $V_{IN}$ is reduced to $-0.8$ V, $\phi_{0D}$ is decreased, and the potential difference between 0D interface and source is reversed by negative $\phi_{0D}$, so negative $I_S$ is exhibited (light red lines in Fig. 2f). When $V_{IN}$ is increased, the shunt current is reduced and the difference between $\phi_{0D}$ and $V_S$ is decreased, so the absolute of negative $I_S$ is reduced and eventually reach to zero. With the increase of $V_{CG}$, electron carrier flow from 0D interface to drain is enhanced by lowered band of the CG-controlled channel, $\phi_{0D}$ approaches to or exceeds $V_S$, and the $I_S - V_{IN}$ transport curve is lifted (from light to dark red lines in Fig. 2f). For $V_{CG} > -0.60$ V, $\phi_{0D}$ is close to the drain voltage $V_D$, so the curve of $I_S$ approximates to the transfer curve of w/o-filament state. The energy band variations with different voltage configurations are shown in Supplementary Fig. 6. The saturation characteristics in w/o-filament state, bidirectional current modulations, and current approximation effect jointly enable the functionality and stability of attention distribution computing. Figure 2g illustrates attention distribution computing mechanism, where $V_{IN-}$ and $V_{IN+}$ represent input voltage levels of logic 0 and 1, respectively. Transport curves in different states vary with $V_{CG}$. Spatial attention ($\alpha_{spatial}$) and temporal attention ($\alpha_{temporal}$) are used to quantify the intensities of spatial and temporal information in the output, which are given by:

$$\alpha_{spatial} = \frac{I_S(V_{IN} = V_{IN+}, w/-filament)}{I_{S0}} \quad (1)$$

$$\alpha_{temporal} = \frac{I_S(V_{IN} = V_{IN-}, w/o-filament) - I_S(V_{IN} = V_{IN-}, w/-filament)}{I_{S0}} \quad (2)$$

where $I_{S0}$ denotes the maximum source current under a given $V_{CG}$. With the increase of $V_{CG}$, $\alpha_{spatial}$ is increased, and $\alpha_{temporal}$ is decreased (Fig. 2h), which enables dynamic adjustment of attention.

For determination computing, filament states store weight information, and the determination signal is mapped from situation values. Drain and source are reversely biased. The attention-inspired device exhibits bidirectional $I_S$ responses, which enable bipolar weighted analog computing (Fig. 2i). In w/o-filament state, 2D homojunction is formed by the gating of $V_{IN}$. For negative $V_{IN}$ ($V_{IN} < -0.50$ V), the homojunction potential barrier is reversely biased to cut off the channel. For positive $V_{IN}$ ($V_{IN} > 0.20$ V), the barrier is forwardly biased, and $I_S$ is turned on (the blue line in Fig. 2i). In w/-filament state, when $V_{IN} < -0.50$ V, negative shunt current is imported from 0D interface, and $\phi_{0D}$ is reduced. A large $V_{CG}$ (0.6 V) bends down the CG-controlled channel and increases electron carrier density from 0D interface to drain, which attenuates the potential reduction effect of $\phi_{0D}$ induced by shunt current. For $V_{IN} < -0.50$ V, the homojunction potential barrier is reversely biased, and $I_S$ is reduced to the sub-threshold level. When $V_{IN} > 1.30$ V, positive shunt currents are induced, $\phi_{0D}$ is larger than $V_S$, and $I_S$ is positive (the red line in Fig. 2i). Determination computing is performed by:

$$I_S = I_e \cdot (w \cdot x_{IN}) \quad (3)$$

where $I_e$ is denoted as the unit current, and $w$ is the weight. $w = w_-$ in w/o-filament state and $w = w_+$ in w/-filament state. $x_{IN}$ denotes the input data logic ($x_{IN} = 0$ for $V_{IN} = V_{IN-}$ and $x_{IN} = 1$ for $V_{IN} = V_{IN+}$). When $x_{IN} = 0$,

the attention-inspired device is resting ($I_S$ is zero). When $x_{IN} = 1$, the attention-inspired device exhibits excitatory ($I_S$ is positive, w/-filament state) or inhibitory ($I_S$ is negative, w/o-filament state) behaviors. The weight plasticity property of the attention-inspired device is shown in Fig. 2j, where the ratio of $w_+$ and $w_-$ can be linearly adjusted from 0 to 1 by $V_{IN+}$. Shunt current characterizations of the attention-inspired device corresponding to the functionalities in computing mode are provided in Supplementary Note 3. The switching between writing and computing modes is repeatable, and the 2D electrostatic modulation is stable after multiple filament state transfer cycles (Supplementary Note 4).

## Hardware adaptive spatial-temporal information processing

Functionalities of the attention-inspired device can be applied to adaptive spatial-temporal information processing. Figure 3a demonstrates an adaptive spatial-temporal information processing primitive based on attention-inspired device arrays. The array circuit schematic is shown in Supplementary Fig. 9. Situations are input to the determination network to determine the optimized attention (Fig. 3b). $I_{det}$ denotes the determination current that linearly maps to the spatial attention. When spatial info. demand is true (T) and temporal info. demand is false (F), $I_{det}$ is positively large (2.99 μA), spatial attention is higher than temporal attention, and spatial information is focused. When there is temporal but not spatial info. demand, $I_{det}$ is negatively large ($-2.81$ μA), temporal attention is higher than spatial attention, and temporal information is focused. When both information is demanded or both is not demanded, $I_{det}$ is close to 0, and both spatial and temporal information is obtained. $I_{det}$ values in different situations are listed in Supplementary Table 2. A data stream with 5×5 units captures a moving object A and static object B (Fig. 3c), and is input to the attention-inspired device array. Data of each input frame is shown in Supplementary Fig. 10. A single frame is output from the array, containing adjustable spatial and temporal information under given attention distributions. Figure 3d illustrates the information processing flow. Frames of the data stream are sequentially input and stored. For each data update period, the input data ($t_2$) interacts with the stored data ($t_1$) and outputs the frame with enhanced spatial or temporal information, then the input data is stored in the array and will be interacted with the next input data in the following period. The output flow at $t_1$ (Fig. 3e, h, k), $t_2$ (Fig. 3f, i, l), and $t_3$ (Fig. 3g, j, m) are obtained. When spatial information is focused (spatial attention is 100%, Fig. 3e–g), locations of object A and B at each time are shown in output frames, and temporal information is neglected. When complete information is focused (spatial attention is 40%, Fig. 3h–j), complete spatial and temporal information is extracted. The moving direction of A is detected, and locations of both A (moving) and B (static) are obtained. When temporal information is focused (spatial attention is 0%, Fig. 3l–m), the motion feature of A is largely highlighted and the static object B is neglected. Output source data matrices are provided in Supplementary Fig. 11. The current values less than 10 nA are under the noise level. The statistical analysis of the fabricated attention-inspired devices is shown in Supplementary Fig. 12.

## Attention-enhanced high-efficiency edge intelligence

The attention-inspired device can be used for edge intelligence scenarios that require immediate processing with reduced hardware overhead. Attention-enhanced autonomous driving platforms are demonstrated to reveal the adaptability of the attention-inspired device to perform full-range attention distribution and real-time response to dynamic situation variations for ever-changing environments. Figure 4a illustrates the output images of the attention-enhanced infrastructure under a full range of attention distribution from 0 to 100%. When spatial attention is 0%, the background and static objects are neglected. Temporal information of moving vehicles and pedestrians is detected. When spatial attention is 100%, the system

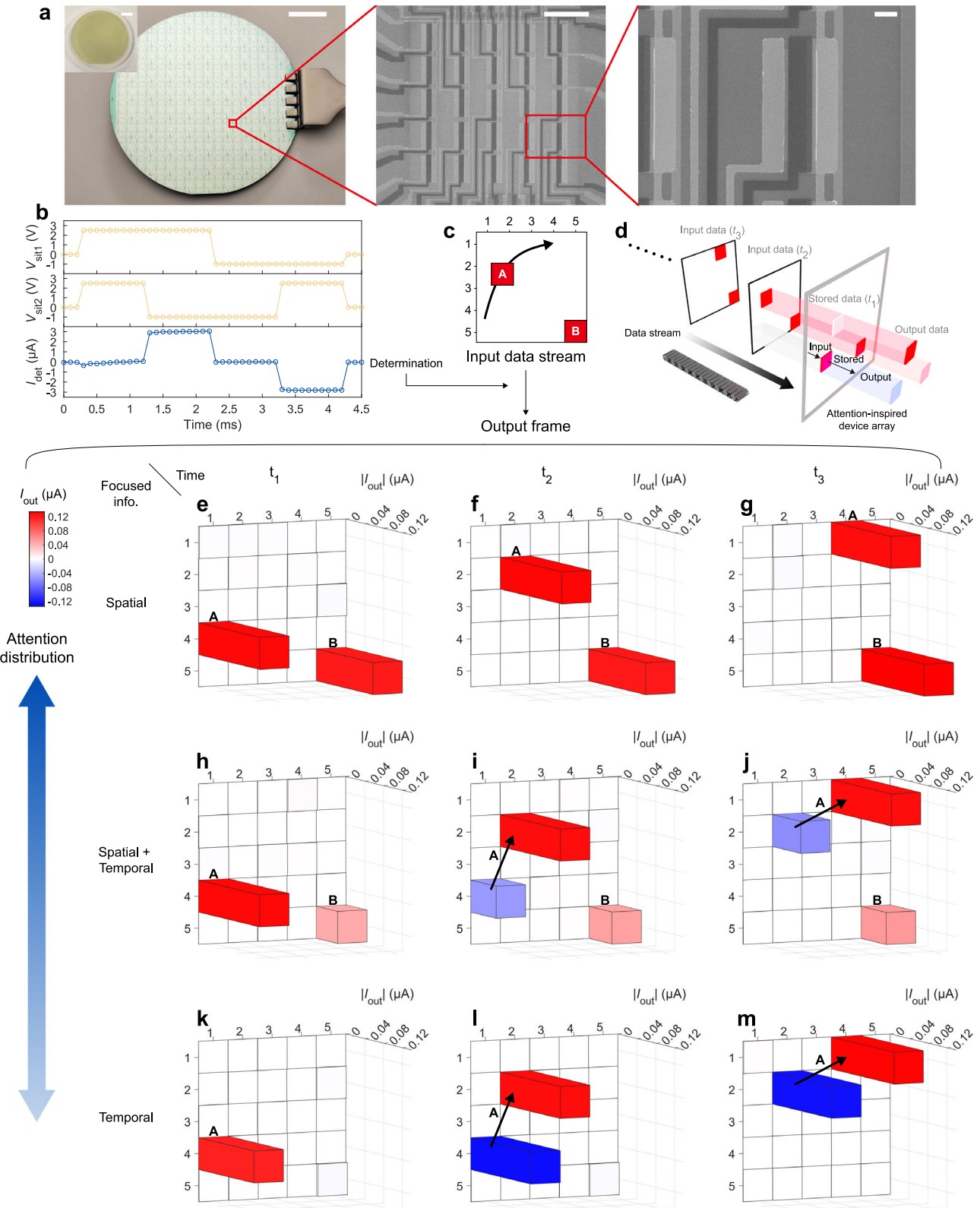

**Fig. 3 | Attention-inspired device arrays for hardware adaptive spatial-temporal information processing. a** Optical image of a 2 inch wafer production. Scale bar, 1 cm. Inset: CVD Monolayer $MoS_2$ on sapphire. Scale bar, 1 cm. Scanning electron microscopy image of a 5 × 5 attention-inspired device array (Scale bar, 100 μm) and each device unit (Scale bar, 10 μm). **b**, Experimental results of determination network. $V_{sit1}$ represents the situation 1 (spatial info. demand) voltage. $V_{sit2}$ represents the situation 2 (temporal info. demand) voltage. T (logic 0) means that the situation is true, and F (logic 1) means that the situation is false, which are represented by $V_{sit-} = -1.00$ V and $V_{sit+} = 2.50$ V respectively. **c** Schematic of the input data stream that captures a moving object A (the arrow marks the moving direction), and a static object B. **d** Data processing flow of the attention-inspired device array. Each frame is sequentially input and stored in the array. (**e–g**) Outputs in $t_1$ (**e**), $t_2$ (**f**), and $t_3$ (**g**) contain spatial information and neglect temporal information. **h–j** Outputs in $t_1$ (**h**), $t_2$ (**i**), and $t_3$ (**j**) process complete spatial and temporal information. **k–m** Outputs in $t_1$ (**k**), $t_2$ (**l**), and $t_3$ (**m**) focus on temporal information.

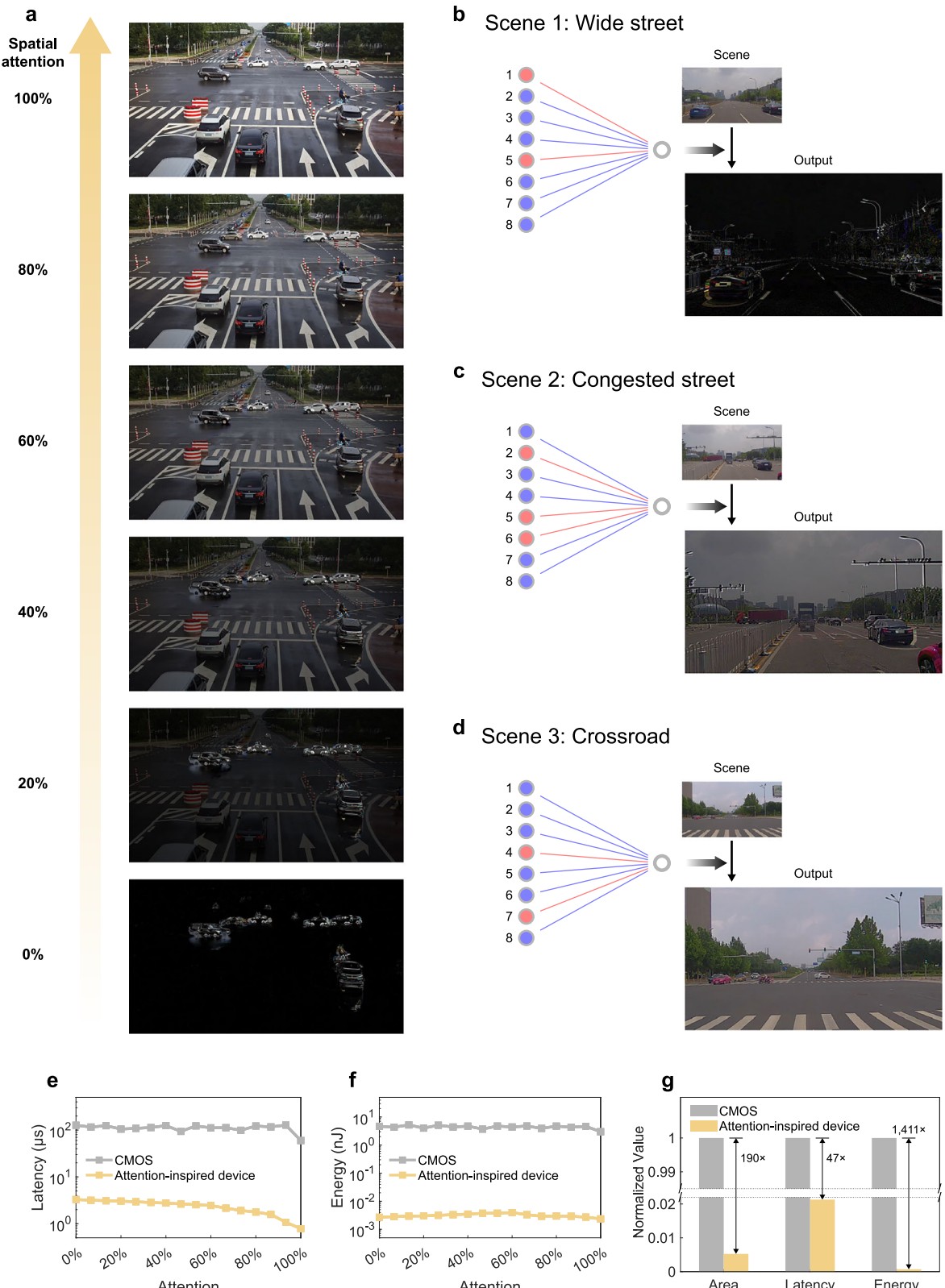

**Fig. 4 | Highly adaptive edge equipment and performance projections.**
**a** Illustrations of the attention-enhanced infrastructure. Attention can be continuously adjusted from 0 to 100%, and examples under 6 values of spatial attention are shown in (**a**). (**b**–**d**) Illustrations of the attention-enhanced vehicle. Attentions are dynamically adjusted in different scenes, including wide street (**b**), congested street (**c**), and crossroad (**d**). **e**–**f** Simulated latency (**e**) and energy (**f**) projections of the conventional transistor-based circuit and the attention-inspired-device-based architecture for spatial-temporal information processing with attention adjustment from 0 to 100%. **g** Area, latency, and energy improvements of the attention-inspired device.

exclusively detects spatial information. The output image contains all the objects, and does not extract temporal information. When spatial attention is not 0 or 100%, complete spatial and temporal information is detected by the attention-enhanced infrastructure. With the increase of spatial attention, the system draws more attention to spatial information and less to temporal information. Attention is adjustable in real-time to ensure dynamic adaptation to varying traffic situations. Detailed implementation processes are provided in Supplementary Note 5. Besides, attention-enhanced equipment is compatible with present artificial intelligence algorithms for complex information recognition tasks (Supplementary Note 6).

Attention-enhanced vehicle is demonstrated with dynamic responses to situation variations (Supplementary Table 5) in different scenes. When the vehicle is driving fastly on a wide street, temporal information of adjacent vehicles should be emphasized to avoid collision (Fig. 4b). Therefore, highlighted temporal information of adjacent vehicles is required. The attention-enhanced vehicle processes situations and draws major attention to temporal information. Resultantly, moving cars and road lines that are required for routing are highlighted in the output. When the vehicle is driving on a congested street, the movement of relatively fast vehicles should be monitored to remind the system to take care, and locations of slow or static vehicles should be detected as well (Fig. 4c). Spatial and temporal information should be captured with optimized proportion. The vehicle analyzes situations and outputs both spatial and temporal information, where the fast black car at the lower right side is highlighted and other vehicles with low speeds are located. When the vehicle is at a crossroad, static objects including traffic lights should be detected (Fig. 4d). Responding to situations, the vehicle attaches major attention to spatial information. The red left-turn signal and green straight-through signal are shown in the output. Attention-enhanced equipment has been verified with the capacity to capture significant information in various scenes, and has the potential to be applied to complicated and ever-changing environments.

For the attention-inspired device, both spatial and temporal information are in situ processed, whereby a large amount of computation resources is saved by the integrated multidimensional information processing functionality. Furthermore, temporal information is sequentially stored in the attention-inspired device, saving the peripheral memory units and data transmission operations. The proposed architecture performance is analyzed and illustrated in Fig. 4e–g. We have compared the proposed architecture to a standard complementary metal-oxide-semiconductor (CMOS) circuit composed of conventional transistors. Verilog-A models of the transistors and schematic circuits have been built, which perform 4-bit attention adjustment of spatial and temporal information with 6.7% attention precision (Supplementary Note 7). The time latencies (Fig. 4e) and energy costs (Fig. 4f) are measured with a variety of attention adjustments from 0% to 100%. The energy costs of peripheral memory units are not included in the total energy consumption of the transistor-based circuit. The attention-inspired device exhibits μs-level time latencies that are more than tenfold lower than the transistor, and maintains pJ-level energy consumption, achieving an energy reduction of three orders of magnitude. The area efficiency, average latency, and average energy are benchmarked (Fig. 4g). The attention-inspired device shows a 190-fold area reduction and a 47-fold latency decrease. Due to the largely reduced number of devices in the circuit, and the highly shortened propagation delay of each operation, 1411-fold energy reduction is achieved.

## Discussion

In conclusion, 0D-2D hetero-dimensional modulations perform in situ attention-inspired information processing, and the modulation strength can be continuously adjusted in a large range. The attention-inspired device based on adjustable hetero-dimensional modulations realizes complete information perception, and has been utilized to establish the adaptive spatial-temporal information processing architecture. Attention-inspired device arrays have been used to process a $5 \times 5$-unit data stream. The optimized attention is adjusted with situation variations in real-time. Experiments of attention distribution from 0 to 100% show a full-range adjustment of spatial-temporal information intensities. The attention-inspired device has been applied to autonomous driving platforms. The attention-enhanced infrastructure and vehicle exhibit dynamic response capability to traffic scene variations. We have benchmarked the performance of the attention-inspired device with 99.5% area, 97.9% latency, and 99.9% energy reductions compared to the transistor-based CMOS circuit. We believe that the proposed attention-inspired architecture can lead to advances in spatial-temporal information perception for edge computing applications.

## Methods

### Transfer of monolayer MoS$_2$

2-inch CVD Monolayer MoS$_2$ on sapphire substrate was bought from Shenzhen 6Carbon Technology Co., Ltd. 8 wt% Poly(methyl methacrylate) (PMMA) solution was spin-coated at 1,500 rpm for 60 s on sapphire/MoS$_2$ and baked at 100 °C for 120 s. After that, the MoS$_2$/PMMA stack was lifted off by deionized water and transferred to the target sample. Then, the excess moisture was air-dried at room temperature for more than 12 h, and an annealing process at 60 °C for 2 h was conducted to remove residues and ensure adhesion between MoS$_2$ and the substrate. The PMMA was removed by acetone for 1 h twice.

### Device fabrication

The SiO$_2$ (300 nm)/p++ Si substrate was patterned by a normal lithography process using AZ601 as the photoresist and wet etching process in a buffered oxide etch (BOE) solution, followed with 15 nm HfO$_2$ layer deposition by atomic layer deposition (ALD). CG electrodes were fabricated through normal lithography using NR9-1000py as the photoresist followed by e-beam evaporation (EBE) of 1 nm Cr/15 nm Pd, and 15 nm ALD-grown HfO$_2$ was deposited. The CVD monolayer MoS$_2$ was wet-transferred on the substrate. The MoS$_2$ pattern was defined by normal lithography using AZ601, and was etched by oxygen plasma treatment with O$_2$ at 150 sccm and 150 W power for 7 min. Then, the drain and source electrodes were fabricated with normal lithography using NR9-1000py and EBE deposition of 5 nm Cr/35 nm Pd. 14 nm HfO$_2$ was deposited by ALD to form the interfacial dielectric. Finally, normal lithography with NR9-1000py and EBE process was conducted to pattern and deposit the 50 nm Ag electrode. ALD processes are carried out at 200 °C.

### Device characterization

The SEM images of devices were measured by Zeiss GeminiSEM 300 (Carl Zeiss, Germany) by Inlens secondary and backscatter electron imaging. The accelerating voltages were 7 kV (the middle image of Fig. 3a) and 10 kV (Fig. 1b, the right side image of Fig. 3a). TEM and EDS images were characterized by Talos F200S G2 S/TEM (Thermo Fisher Scientific). The accelerating voltage was 200 kV. Electrical characterizations were performed on a probe station (TS2000-HP, MPI) connecting to an Agilent B1500A semiconductor parameter analyzer. Pulse measurements were conducted using Keysight B1530A waveform generator/fast measurement units (WGFMUs). Electrical tests were carried out at room temperature and ambient environment.

## Data availability

Image data used for demonstrations is preprocessed from the publicly available dataset[42]. Other data supporting the key findings of this study are provided in the article and the Supplementary Information file. Raw data of the current study are available from the corresponding authors via email.

## Code availability

The codes for plotting the data are available from the corresponding author via email.

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

## Acknowledgements

This work was supported by the National Key R&D Program (2022YFB3204100, 2021YFC3002200), the National Natural Science Foundation (U20A20168, No. 62374099) of China, the Young Scientists Fund of the National Natural Science Foundation of China (No.

62404121), STI 2030—Major Projects under Grant 2022ZD0209200, Beijing Natural Science Foundation-Xiaomi Innovation Joint Fund (L233009) and Beijing Natural Science Foundation (L248104), the Research Fund from Tsinghua University Initiative Scientific Research Program, a grant from the Guoqiang Institute, Tsinghua University, Independent Research Program of School of Integrated Circuits, Tsinghua University, Tsinghua University Fuzhou Data Technology Joint Research Institute, and CIE-Tencent Robotics X Rhino-Bird Focused Research Program.

## Author contributions

F.W. and J.P. conceived the idea and the project. T.-L.R., D.Y., Y.Y., and H.T. supervised the project. J.P., F.W., and Y.L. conducted the experiments. J.P., K.Q., and K.J. performed the simulations. Z.W., P.G., and J.Y. were involved in device fabrication and characterization. J.P. and F.W. co-wrote the manuscript with inputs from all the co-authors. All the authors discussed the results and commented on the manuscript.

## Competing interests

The authors declare no competing interests.
