## [Transparent Peer Review file · Nature Communications]

Adaptive spatial-temporal information processing based on in-memory attention-inspired devices

Corresponding Author: Dr Tian-ling Ren

Version 0:

Reviewer comments:

Reviewer #1

(Remarks to the Author)

In this work, the authors propose an attention-inspired device and strategy designed to emulate the brain's attention mechanism, enabling adaptive spatial-temporal information processing. This adaptability is achieved through a synergistic modulation of metal filament formation and electrostatic control. The device performs both attention distribution and decision-making functions, making it suitable for edge intelligence applications that demand rapid processing with minimal hardware complexity. While the concept shows promise, several critical issues require further clarification and enhancement:

- 1. Device Configuration and Ion Migration:** The multi-terminal device configuration deviates from conventional designs. A schematic diagram should be provided to clearly illustrate the measurement setup. Additionally, as the device operates through Ag filament formation and rupture, it is important to address whether Ag atoms migrate into or along the MoS₂ channel during switching. Such migration could lead to long-term degradation or failure, and this potential risk should be discussed.
- 2. Switching Endurance and Stability:** The current data only show filament state transitions over three cycles (Supplementary Fig. 1), which is insufficient to demonstrate device stability. Extended endurance testing is necessary to assess reliability. Furthermore, the authors should include a discussion on degradation mechanisms that could affect device performance over repeated switching cycles.
- 3. Multi-level Conductance and Retention:** Since emulating attention requires distinct non-volatile filament states, the device's ability to exhibit multi-level conductance modulation through electrical pulses, along with long-term retention across conductance states, should be experimentally demonstrated.
- 4. Terminology Clarity:** The meaning of "situation" in the sentence "The unit is resting when the situation is false, and is excitatory (with filament) or inhibitory (without filament) when the situation is true" (Lines 104–106) is ambiguous. The authors should define what constitutes a "true" or "false" situation within the device's operating context.
- 5. Energy Band Diagrams:** The working principles discussed (Lines 153–188), involving barrier modulation and band bending, would benefit greatly from illustrative energy band diagrams to improve intuitive understanding of the mechanisms.
- 6. Wafer-Scale Integration Details:** The claim in Line 232–233 regarding "wafer-scale fabrication of an adaptive spatial-temporal information processing system" lacks specificity. The term "system" should be clarified—does it refer to full-wafer integration or a small array on a 2-inch wafer? Additional details on device density, yield, and variation across the wafer are needed to assess scalability.
- 7. Comparison with CMOS:** The reported improvements over CMOS—190× area, 47× latency, and 1411× energy savings—lack substantiation. The specific CMOS baseline (architecture and transistor type) is not described, and supporting references or data are absent. Furthermore, the evaluation methodology for area, latency, and energy must be detailed to ensure the comparison is fair and technically valid.
- 8. Implementation Details:** The implementation of the attention-enhanced edge intelligence application is unclear. If circuit-based, the authors should provide the circuit schematics; if based on software simulation, detailed simulation methodology

and a flowchart should be included to explain the operation pipeline.

Minor Issue:

9. In Figure 1, the panel labels in the caption are incorrect—'a' and 'b' are mislabeled as 'd' and 'e' (Lines 114 and 116). Please correct these labeling errors.

Reviewer #2

(Remarks to the Author)

The authors develop an attention-inspired device structure based on two-dimensional MoS₂ channel and non-volatile memory behavior of zero-dimensional Ag filament interfaces, analyze the physical fundamentals of the hetero-dimensional modulation mechanism in the device, and show the adaptive spatial and temporal information processing capability of the device, which is demonstrated to be used for highly efficient edge intelligence computing in dynamic scenarios. This paper provides a novel analog computing-in-memory architecture, other than the traditional crossbar-based computing-in-memory that is specifically designed for matrix-vector multiplications. The former has outstanding performance in dynamic information processing. This work would be of interest to researchers focusing on artificial intelligence hardware development with novel materials and device structures. Therefore, I think the manuscript has the potential to be accepted, but a few problems should be addressed before acceptance.

1. Several data about the device's fundamental properties are missing. Although the transport and output behavior curves of the attention-inspired device have been shown, it is recommended to add more data about the MoS₂ channel and filament properties respectively, such as the transfer curve of the MoS₂ transistor, and resistance switching property of the filament.
2. The non-volatile memory property is shown in Fig. 2b. Is the non-volatile memory behavior due to the hysteresis of the MoS₂ channel? The MoS₂ channel hysteresis should be added.
3. It is suggested to give the circuit schematic of the wafer scale 5×5 attention-inspired device array in Fig. 3a.
4. Since there are simulations in Section 4, it is unclear about the simulation process conducted in this section. The point is how to use the electric properties of the fabricated device to construct a system for edge intelligence.
5. The curve fitting parameters of Fig. 2h and Fig. 2j are missing. Those data may be critical for the modeling of the device.
6. The attention is adjusted by VCG in the device in the description of the attention-inspired device principle in Section 3. Is the attention in Section 4 also adjusted by VCG? If so, the author should give the specific values of the CG voltages corresponding to all the attention values in Fig. 4a, as well as the adjusted attention values in the three scenes in Fig. 4b-d.

Reviewer #3

(Remarks to the Author)

The manuscript J. Pan et al. reports on an attention-inspired artificial intelligence Architecture, using the functionalities of Ag(HfO₂/MoS₂) memristive devices and highlight the efficiency compared to conventional transistor based circuits. The manuscript presents interesting results, and the topic is definitely of interest for the journal. However, there are several issues that need to be addressed prior to a decision on the manuscript.

Comments:

Whereas I do not have concerns about the computational part, there are several issues explaining the device the authors use.

1. In fact the devices used are typically termed as memristors and more specifically ECM (or CBRAM) memristive devices. This seems to be overlooked in the manuscript.
2. The functions and critical parameters of ECM devices are well known, but missing in the manuscript e.g. endurance, retention, switching time, as well the neuromorphics-related characteristics. It should be clarified how reliable are the devices and compare them to state-of-art memristive devices.
3. It is also well-known that oxide-type ECM memristors are sensitive to water molecules, protons/moisture coming from surroundings and/or during device preparation steps. Moisture effects can significantly alter the device performance. Could the authors comment on this?
4. It should be discussed and clarified, what is the reason (and advantages) on taking the Ag/HfO₂/MoS₂ system. What is the role of the materials composition and thicknesses. For example it should be explained why Ag is selected and not (for example Cu), as well why HfO₂ and not e.g. TaOx or SiO₂, etc.
5. How the selected devices facilitate the observed advanced functionalities.
6. The operation mechanism needs more clear explanation and justification.

The above mentioned points should be implemented in the text supported by corresponding citations from the literature.

Version 1:

Reviewer comments:

Reviewer #1

(Remarks to the Author)

The rebuttal is complete, detailed, and technically sound. Each concern was addressed with new data, clarified explanations, or corrected terminology. In particular, the authors demonstrated substantial responsiveness to criticisms regarding endurance, performance benchmarking, and integration scale. The revisions collectively strengthen the manuscript's rigor and presentation.

Reviewer #2

(Remarks to the Author)

The concerns have been addressed in the revised manuscript and SI. The current version of the manuscript could be accepted.

Reviewer #3

(Remarks to the Author)

The authors have addressed the comments and revised/amended the manuscript. In my opinion the work can be accepted for publication.

REVIEWER COMMENTS

Reviewer #1 (Remarks to the Author):

In this work, the authors propose an attention-inspired device and strategy designed to emulate the brain's attention mechanism, enabling adaptive spatial-temporal information processing. This adaptability is achieved through a synergistic modulation of metal filament formation and electrostatic control. The device performs both attention distribution and decision-making functions, making it suitable for edge intelligence applications that demand rapid processing with minimal hardware complexity. While the concept shows promise, several critical issues require further clarification and enhancement:

Response:

We sincerely thank the reviewer for the positive comments on our manuscript and the constructive comments that help us to improve the quality and integrity of this work. We have added required experiments about the device's retention, endurance, yield and variation, provided analyses of the Ag filament stability, and clarified the expressions and methodologies about measurement setup, working principle, terminologies, performance comparison, and edge intelligence implementation according to your requirements. We think this response can solve all your questions and requests. The detailed point-to-point responses are given as follows.

1. Device Configuration and Ion Migration: The multi-terminal device configuration deviates from conventional designs. A schematic diagram should be provided to clearly illustrate the measurement setup. Additionally, as the device operates through Ag filament formation and rupture, it is important to address whether Ag atoms migrate into or along the MoS₂ channel during switching. Such migration could lead to long-term degradation or failure, and this potential risk should be discussed.

Response:

The reviewer has raised important points about the working mechanisms of the device. For the device configuration, we have added schematic diagrams illustrating the measurement

setups in the revised version. The measurement setup schematic is shown in Fig. R2e.

Figure. R2e, Structure of an attention-inspired device working in writing and computing modes.

The Ag electrode (IN) is connected to the input data stream, and the control gate (CG) continuously adjusts the output behavior. In writing mode, the CG voltage V_{CG} controls the on-state current by buried gate electrostatic modulation of the MoS₂ channel. V_{IN} is applied to IN, and the drain voltage V_D is grounded. In computing mode, the source and substrate terminals are connected to voltages V_S and V_B , and V_{CG} is continuously tunable to adjust the attention. According to your request, we have also added descriptions of measurement setup in the main text:

Lines 130–136:

The Ag electrode (IN) is connected to the input data stream, and the control gate (CG) continuously adjusts the 0D-2D hetero-dimensional modulation characteristics (Fig. 2e). In writing mode, the input voltage V_{IN} is applied to IN, and the drain voltage V_D is grounded. The CG voltage V_{CG} controls the on-state current by buried gate electrostatic modulation of the MoS₂ channel. 0D contact interface between MoS₂ and Ag filament is formed by applying positive writing voltage V_{IN} , and is ruptured by negative V_{IN} (Fig. 2c).

Lines 164–166:

In computing mode, the attention-inspired device exhibits reconfigurable properties to implement attention distribution and determination computing modes by the source voltage V_S and the CG voltage V_{CG} . V_{CG} is tunable to adjust the attention.

According to previous studies, there are van der Waals gaps between Ag and MoS₂

interfaces (*ACS Appl. Mater. Interfaces* **2021** 13 (13), 15802-15810), and Ag atoms do not migrate into the channel of 2D TMD layer during switching (*ACS Nano* **2019** 13 (2), 2205-2212). Based on the result, the theoretical model on the Fermi-level unpinning effect of the contact between Ag filament and 2D TMD layer was proposed recently (*ACS Nano* **2024** 18 (43), 29771-29778), which accords with the experimental observation that the Fermi-level pinning factor is increased from 0.06 (the contact between bulk Ag and 2D TMD) to 0.95 (the contact between Ag filament and 2D TMD). Since there are no migrations into MoS₂ layer, Ag atoms are fixed by the HfO₂ oxide layer and do not move along the layer. Ag was not observed along the MoS₂ layer during our TEM characterization. Therefore, we think it is probable that Ag atoms do not migrate into or along the MoS₂ channel during switching, and degradation and failure are not induced by the interface. The oxide film degradation is the main reason for the device failure, which is the same as other resistive switching devices.

2. Switching Endurance and Stability: The current data only show filament state transitions over three cycles (Supplementary Fig. 1), which is insufficient to demonstrate device stability. Extended endurance testing is necessary to assess reliability. Furthermore, the authors should include a discussion on degradation mechanisms that could affect device performance over repeated switching cycles.

Response:

Thank you for pointing out the insufficient data in the previous manuscript. As the increase of the switching cycles, defects are introduced to the oxide film. When the defect density gradually becomes large, a conductive path would be formed through the dielectric, and the dielectric would be broken down, leading to the switching function degradation. We have added the experiment of endurance testing, as shown in Supplementary Fig. 5b in the revised version. The device can be continuously transferred between two states without degradation for up to 100 cycles. The proposed device structure is mainly to demonstrate a design to achieve in situ spatial-temporal information processing. To validate our design functionalities, Ag was chosen as the top electrode due to its fabrication viability in our laboratory, and the proposed structure

can also be realized by other resistive-switching materials. Although Ag-based resistive switching devices have relatively lower endurance than VCM including TiN-based devices, several strategies have been reported to enlarge the endurance, including adding a stacking layer structure⁷, widening the intrinsic threading dislocation that acts as the preferential diffusion paths of Ag in the dielectric⁸, reducing the quantity and movement of metal ions during switching via partial filament formation and variations⁹, etc. The discussions of degradation are included in Supplementary Note 4.

Supplementary Figure R5b, Endurance testing.

References:

7. Yan, X. et al. Robust Ag/ZrO₂/WS₂/Pt Memristor for Neuromorphic Computing. *ACS Appl. Mater. Interfaces* **11**, 48029-48038 (2019). <https://doi.org/10.1021/acsami.9b17160>
8. Choi, S. et al. SiGe epitaxial memory for neuromorphic computing with reproducible high performance based on engineered dislocations. *Nat. Mater.* **17**, 335-340 (2018). <https://doi.org/10.1038/s41563-017-0001-5>
9. Kim, J., Kwon, O., Seo, J. & Hwang, H. Vertical-Switching Conductive Bridge Random Access Memory with Adjustable Tunnel Gap and Improved Switching Uniformity Using 2D Electron Gas. *Adv. Electron. Mater.* (2024). <https://doi.org/10.1002/aelm.202400650>

3. Multi-level Conductance and Retention: Since emulating attention requires distinct non-volatile filament states, the device's ability to exhibit multi-level conductance modulation through electrical pulses, along with long-term retention across conductance states, should be experimentally demonstrated.

Response:

Thanks for your question. The attention adjustment emulation in this work is achieved by tuning the voltage level of V_{CG} , not by multi-level conductance. The weighted mappings between the input, stored, and output data are established by the transport behavior that can be continuously adjusted by the V_{CG} (Fig. 2f). With the increase of the V_{CG} , spatial information is increased, and temporal information is decreased (Fig. 2g). The spatial and temporal attention are continuously adjusted from 0% to 100% or from 100% to 0% with the increase of V_{CG} (Fig. 2h).

Figure 2 | Working mechanisms of the attention-inspired device. f, I_S - V_{IN} transport curves under different V_{CG} from -1.80 to 0.00 V. Device structure schematics illustrate states of the attention-inspired device and current directions under each voltage configuration. IN and CG electrode voltage configurations are represented by blue (low voltage), red (high voltage), or gray (arbitrary voltage) colors. **g**, I_S - V_{CG} attention distribution characteristics. $V_{IN-} = -0.72$ V,

$V_{IN+} = -0.36$ V. **h**, Spatial and temporal attention varying with V_{CG} that exhibit dynamic attention adjustment properties. Dash lines show the exponential fitting.

The previous manuscript did not clearly explain the functionality, and the relevant discussion has been clarified. Furthermore, retention testing has been conducted and shown in Supplementary Fig. R5a.

Supplementary Figure R5a, Retention testing.

Revisions in the main text:

Lines 164–169:

In computing mode, the attention-inspired device exhibits reconfigurable properties to implement attention distribution and determination computing modes by the source voltage V_S and the CG voltage V_{CG} . V_{CG} is tunable to adjust the attention. For attention distribution computing, transport curves varying with V_{CG} in different filament states to emulate attention-directed perception in the brain are illustrated in Fig. 2f (positive direction of I_S is from drain to source).

Lines 198–200:

With the increase of V_{CG} , α_{spatial} is increased, and α_{temporal} is decreased (Fig. 2h), which enables dynamic adjustment of attention.

4. Terminology Clarity: The meaning of “situation” in the sentence “The unit is resting when the situation is false, and is excitatory (with filament) or inhibitory (without filament) when the situation is true” (Lines 104–106) is ambiguous. The authors should define what constitutes a “true” or “false” situation within the device’s operating context.

Response:

We apologize for the missing terminology definition. We have revised the discussion and clarified the terminology. “True” means that the situation description matches with the present scenario, and “False” means that the situation description does not match with the present scenario. Therefore, situation information can be encoded as logic signals input in the network.

In Lines 105–109:

Situations are encoded as logic signals and input to the network. Situation descriptions include “the vehicle has a high speed”, “there is heavy traffic on the road”, etc. “True” (logic 1) or “False” (logic 0) indicates whether the situation description is real. The unit is resting when the situation is false, and is excitatory (w/-filament state) or inhibitory (w/o-filament state) when the situation is true.

5. Energy Band Diagrams: The working principles discussed (Lines 153–188), involving barrier modulation and band bending, would benefit greatly from illustrative energy band diagrams to improve intuitive understanding of the mechanisms.

Response:

The reviewer has pointed out an excellent approach to clearly explain the working principles of the proposed device by energy band diagrams. We have added energy band diagrams to Supplementary Fig. 6, where the diagrams illustrate energy band states under each voltage configuration. The energy band diagrams correspond to the device structure schematics and transport behavior curves in Fig. 2f.

Supplementary Figure R6. Energy band diagrams of the voltage configurations in attention distribution computing. “S” denotes the source terminal, “D” denotes the drain terminal, and “0D” denotes the 0D interface.

Figure 2 | Working mechanisms of the attention-inspired device. f, I_S – V_{IN} transport curves under different V_{CG} from –1.80 to 0.00 V. Device structure schematics illustrate states of the attention-inspired device and current directions under each voltage configuration. V_{IN} and V_{CG} electrode voltage configurations are represented by blue (low voltage), red (high voltage), or gray (arbitrary voltage) colors.

6. Wafer-Scale Integration Details: The claim in Line 232–233 regarding “wafer-scale fabrication of an adaptive spatial-temporal information processing system” lacks specificity.

The term “system” should be clarified—does it refer to full-wafer integration or a small array on a 2-inch wafer? Additional details on device density, yield, and variation across the wafer are needed to assess scalability.

Response:

The descriptions of the previous manuscript were unclear. We have developed 2-inch wafer fabrication of the attention-inspired device arrays. It is feasible to achieve full-wafer integration of the proposed architecture. Considering the complexity of electrical measurement, we demonstrated the functionality of a 5×5 device array. We agree with the reviewer that the terms “wafer-scale fabrication” and “system” may be inaccurate in describing the results of this work. In the revised version, we have corrected the relevant explanations, and the term “system” is replaced by “primitive” to clarify that the fabricated sample is composed of small arrays to validate the feasibility of the proposed architecture:

Lines 26–29:

To demonstrate the proposed architecture feasibility, an adaptive spatial-temporal information processing primitive is successfully implemented. Experiments of attention adjustments in response to different situations validate the adaptation capability to environmental changes.

Lines 68–70:

An adaptive spatial-temporal information processing architecture has been implemented based on reconfigurable properties of the attention-inspired device to perform attention distribution and determination computing functionalities.

Lines 89–91:

The attention-inspired device implements in-memory analog spatial-temporal computing based on interactive modulations of the 0D-2D hetero-dimensional interfaces. An attention-inspired adaptive spatial-temporal information processing architecture is illustrated in Fig. 1c.

Figure 1 caption:

Figure 1 | Architecture of the adaptive spatial-temporal information processing based on attention-inspired devices. c, Schematic of the adaptive spatial-temporal information processing architecture.

Lines 244–245:

Figure 3a demonstrates the fabricated adaptive spatial-temporal information processing primitive based on attention-inspired device arrays.

Device density, yield, and variation analyses:

The proposed device integrates both spatial and temporal information processing into the device level, thereby saving a large amount of device numbers needed for time-variant information processing. To implement spatial-temporal information processing based on conventional CMOS circuits, 5 NAND gates, 5 AND gates, and 5 full adders are required, which are composed of 190 transistors. In contrast with conventional architecture, the operation of each unit is performed by a single attention-inspired device. Therefore, the proposed architecture has the advantage of device density, yet the area of the fabricated sample in this work is limited by the technology node. It is feasible to largely enhance the device density with advanced technology nodes.

To visualize the yield and variation of the proposed device, statistical analyses have been conducted by measuring the output currents of 25 different devices. The statistical results are shown in Supplementary Fig. 12. There are 4 combinations of input and stored data values, and 3 typical attention configurations are measured in each data combination. Among the 25 devices, 24 samples work properly. The yield is 96%. The output variations are less than 20 nA.

Supplementary Figure R12. Statistical analyses of the attention-inspired device. The output currents of 25 devices are measured with different input and stored data, and different attention configurations (spatial, spatial+temporal, and temporal). Among the 25 devices, 24 are functioning properly.

7. Comparison with CMOS: The reported improvements over CMOS—190 \times area, 47 \times latency, and 1411 \times energy savings—lack substantiation. The specific CMOS baseline (architecture and transistor type) is not described, and supporting references or data are absent. Furthermore, the evaluation methodology for area, latency, and energy must be detailed to ensure the comparison is fair and technically valid.

Response:

The reviewer has raised an important issue about the performance projection of the proposed device. The CMOS circuit for performance comparison uses complementary logic gate architecture that is composed of standard single-gate field-effect transistor. The performance analysis process and the evaluation methodology for area, latency, and energy are provided in Supplementary Note 7 in detail.

To analyze the performance of the attention-inspired device, simulation tests of the adaptive spatial-temporal information processing circuit based on the attention-inspired device have been conducted to evaluate the latency and energy cost. A Verilog-A model of the attention-inspired device has been established. The modeling parameters are extracted by the experimental data of transport and output curves. Then a schematic model of the device is encapsulated, and is referenced by the circuit-level simulation project. Pulse signals are input to the device to simulate the input data stream, and attention is adjusted by the CG voltage level. Propagation delay and averaged currents are measured. The value of latency equals the propagation delay, and the energy cost E is given by:

$$E = (|V_{IN} \overline{I_{IN}}| + |V_S \overline{I_S}| + |V_D \overline{I_D}|) \cdot t \quad (S1)$$

where t is the latency, and $\overline{I_{IN}}$, $\overline{I_S}$, $\overline{I_D}$ are averaged current at the IN, source, and drain terminals respectively. A comparison of the attention-inspired-device-based architecture to a standard CMOS circuit has been conducted. The CMOS circuit implementing the equivalent adaptive spatial-temporal information processing functionality needs to realize the following operation:

$$D_{out}(t) = \alpha_{spatial} \cdot D_{in}(t) + \alpha_{temporal} \cdot [D_{in}(t) - D_{in}(t-1)] \quad (S2)$$

where $D_{in}(t)$ and $D_{out}(t)$ are input and output data respectively. Multiple digital arithmetic units are needed to implement the operation. For example, to implement 4-bit spatial and temporal attention adjustment, 5 NAND gates, 5 AND gates, and 5 full adders are required, even not considering the peripheral memory units storing the previous input data. A schematic model of the CMOS circuit is established. Latency and energy are measured by pulse signal tests. The latency and energy measurement results of the proposed architecture and the CMOS circuit at each attention value are shown in Fig. 4e–f. Since the attention-inspired device integrates

multidimensional information processing into the device level, a large amount of device number is saved for spatial-temporal information processing. 4 transistors, 6 transistors, and 28 transistors²⁸ are required for NAND gates, AND gates, full adders respectively. Therefore, area of the CMOS circuit is:

$$A_{\text{CMOS}} = 5 \cdot A_{\text{NAND}} + 5 \cdot A_{\text{AND}} + 5 \cdot A_{\text{adder}} = 190A_0 \quad (\text{S3})$$

where A_0 is the transistor area. The attention-inspired device has the same area as transistor. Since the operation of each unit is performed by a single attention-inspired device, the area is reduced by 190 compared to the CMOS circuit.

References:

28. Rabaey, J. M., Chandrakasan, A. & Nikolić, B. *Digital integrated circuits: a design perspective* Ed. 2 (Pearson, 2002).

8. Implementation Details: The implementation of the attention-enhanced edge intelligence application is unclear. If circuit-based, the authors should provide the circuit schematics; if based on software simulation, detailed simulation methodology and a flowchart should be included to explain the operation pipeline.

Response:

Thank you for your suggestions. The implementation is based on software simulation. A detailed discussion on the implementation of the edge intelligence application has been added to Supplementary Note 6.

The proposed attention-inspired device is modeled based on experimental data. We conduct exponential fitting to obtain an empirical model of spatial and temporal attentions varied with V_{CG} . The model function is $y = a + be^{c(x+d)}$, where x is the value of V_{CG} , and y is the spatial or temporal attention. Then an empirical model of the maximum source current I_{S0} is built, as shown in Supplementary Fig. R13. The parameter values are listed in Supplementary Table R3. Mappings between V_{CG} , V_{IN} , and I_{S} are established by the models and Eq. (1) of the text.

The flow chart of the attention-enhanced edge intelligence implementation is shown in Supplementary Fig. R14. Various situations are considered in different scenes as the inputs of determination computing. “T” and “F” indicate whether the situation matches with the real scene. An $m \times n$ array is built to perform n -bit perception from the input situations. $m = 8$ is the number of situations. n is set to be 4 in this demonstration. The output determination currents $I_{\text{det}1} - I_{\text{det}n}$ are obtained by analog computing:

$$I_{\text{det}j} = I_e \sum_{i=1}^m (w_{i,j} \cdot x_i), \quad j = 1, 2, \dots, n \quad (\text{S1})$$

where $x_i \in \{0, 1\}$ is the logic of the situation i . $i = 1, 2, \dots, m$. The range of weight for each device cell is $w_{i,j} \in \{-1, 1\}$. $I_{\text{det}1} - I_{\text{det}n}$ are input to the post-processing module to obtain the determination output. Circuit designs of post-processing modules for analog computing include circuits composed of transimpedance amplifier (TIA), analog-to-digital converter (ADC), digital shifter, and adder²¹, or circuits composed of TIA, analog summation node, and ADC²². Accumulation of determination currents in bit 1– n is conducted in the post-processing module to obtain the optimized value of attention in the given situations:

$$\alpha_{\text{spatial}} = \frac{\sum_{j=1}^n 2^j I_{\text{det}j}}{I_{\text{det}0}} \quad (\text{S2})$$

where $I_{\text{det}0}$ is the normalization constant. α_{spatial} is clamped to the range [0%, 100%].

Supplementary Figure R13. Modeling of the maximum source current I_{S0} .

Supplementary Table R3. Exponential fitting parameters.

	Parameters			
	a	b	c	d
I_{S0}	0.130278	-0.0141960	-3.1706	1.20822
α_{spatial}	1.06207	-0.85689	-3.3540	1.26730
α_{temporal}	-0.0036477	1.84917	-5.2985	1.43504

Supplementary Figure R14. Flow chart of the attention-enhanced edge intelligence.

References:

21. Jiang, H., Li, W., Huang, S. & Yu, S. A 40nm Analog-Input ADC-Free Compute-in-Memory RRAM Macro with Pulse-Width Modulation between Sub-arrays. in *IEEE Symposium on VLSI Technology and Circuits (VLSI Technology and Circuits)*. 266-267 (IEEE, 2022).
22. Song, W. et al. Programming memristor arrays with arbitrarily high precision for analog computing. *Science* **383**, 903-910 (2024). <https://doi.org/doi:10.1126/science.adi9405>

Minor Issue:

9. In Figure 1, the panel labels in the caption are incorrect—‘a’ and ‘b’ are mislabeled as ‘d’ and ‘e’ (Lines 114 and 116). Please correct these labeling errors.

Response:

Thank you for pointing out this mistake. We have corrected the error, and rechecked the full text to make sure there were no typos and labelling errors.

Lines 118–120:

d-a, Schematic of an attention-inspired device. The highlighted structure implements 0D-2D hetero-dimensional modulations. The gray and black arrows represent 0D and 2D interfacial modulations. **e-b**, Scanning electron microscopy characteristics of a fabricated attention-inspired device (Scale bar, 10 μm) and the corresponding transmission electron microscopy and energy dispersive spectroscopy mapping (Scale bar, 5 nm).

Reviewer #2 (Remarks to the Author):

The authors develop an attention-inspired device structure based on two-dimensional MoS₂ channel and non-volatile memory behavior of zero-dimensional Ag filament interfaces, analyze the physical fundamentals of the hetero-dimensional modulation mechanism in the device, and show the adaptive spatial and temporal information processing capability of the device, which is demonstrated to be used for highly efficient edge intelligence computing in dynamic scenarios. This paper provides a novel analog computing-in-memory architecture, other than the traditional crossbar-based computing-in-memory that is specifically designed for matrix-vector multiplications. The former has outstanding performance in dynamic information processing. This work would be of interest to researchers focusing on artificial intelligence hardware development with novel materials and device structures. Therefore, I think the manuscript has the potential to be accepted, but a few problems should be addressed before acceptance.

Response:

We sincerely thank the reviewer for the careful reading of this manuscript and acknowledgment of our work. We are encouraged to receive your approval of the novelty and meaningfulness of the proposed design. We have carefully read the comments, and addressed the issues that you have pointed out. The point-to-point reply is as follows:

1. Several data about the device's fundamental properties are missing. Although the transport and output behavior curves of the attention-inspired device have been shown, it is recommended to add more data about the MoS₂ channel and filament properties respectively, such as the transfer curve of the MoS₂ transistor, and resistance switching property of the filament.

Response:

The reviewer has mentioned several important issues about the device's fundamental properties. We agree with the reviewer that it is proper to provide information on those properties. In the revised version, the relevant information is added.

The transfer curve of the MoS₂ transistor is shown in Supplementary Figure 1.

Supplementary Figure R1. MoS₂ channel transport behavior of 10 typical devices. $V_D = 1$ V. The channel length is 2 μm .

The additional resistance-switching properties of the filament, including retention and endurance, are shown in Supplementary Figure 5. The tested retention is larger than 10^4 s. We believe that the actual performance is largely higher than the measured value, since the on-state current degradation of the device at 10^4 s is 1.5%. The tested endurance is larger than 100 cycles.

Supplementary Figure R5. (a) Retention and (b) endurance testing. The read voltage is 0.1 V.

2. The non-volatile memory property is shown in Fig. 2b. Is the non-volatile memory behavior due to the hysteresis of the MoS₂ channel? The MoS₂ channel hysteresis should be added.

Response:

We agree with the reviewer that the non-volatile memory behavior of the 0D interface between MoS₂ and Ag filament should be justified. The MoS₂ channel transport behavior with forward and backward scan curves is shown in Supplementary Figure 2 of the revised version. The hysteresis ratio is largely lower than the state transfer current ratio of the proposed device, which verifies the device mechanism.

Supplementary Figure R2. Forward and backward scan curves of MoS₂ channel. $V_D = 1$ V. The channel width is 50 μm , and the channel length is 2 μm . The ratio (1.066) of the forward and backward scans is largely less than the state transfer current ratio (10^9).

3. It is suggested to give the circuit schematic of the wafer scale 5×5 attention-inspired device array in Fig.3a.

Response:

The reviewer has raised an important issue that the information on the fabricated array circuit schematic should be provided. The 5 × 5 array circuit schematic is added to Supplementary Fig. 9 in the revised version. The data stream is input to the circuit as the voltages $V_{IN\ ij}$ ($i, j = 1 \dots 5$) applied to the array. The output current matrix $I_{out\ ij}$ ($i, j = 1 \dots 5$) contains spatial and temporal information of the input, and can be adjusted by V_{CG} , as shown in Fig. 3e–m.

Supplementary Figure 9. Circuit schematic of the 5×5 attention-inspired device array. $V_{IN\ i,j}$ and $I_{out\ i,j}$ represent the input voltage and output current of the (i,j) pixel.

4. Since there are simulations in Section 4, it is unclear about the simulation process conducted in this section. The point is how to use the electric properties of the fabricated device to construct a system for edge intelligence.

Response:

The logic provided by the reviewer has been taken as a reference during the discussion of the simulation methodology. The proposed attention-inspired device is modeled based on experimental data on transport and output behavior. We conduct exponential fitting to obtain an empirical model of spatial and temporal attentions varied with V_{CG} . Then an empirical model of the maximum source current I_{S0} is built, as shown in Supplementary Fig. R13. Mappings between V_{CG} , V_{IN} , and I_S are established by the models and Eq. (1) of the text.

The flow chart of the attention-enhanced edge intelligence implementation is shown in Supplementary Fig. R14. Various situations are considered in different scenes as the inputs of determination computing. ‘‘T’’ and ‘‘F’’ indicate whether the situation matches with the real scene. An $m \times n$ array is built to perform n -bit perception from the input situations. $m = 8$ is the number of situations. n is set to be 4 in this demonstration. The output determination currents $I_{det1} - I_{detn}$ are obtained by analog computing:

$$I_{detj} = I_e \sum_{i=1}^m (w_{i,j} \cdot x_i), \quad j = 1, 2, \dots, n \quad (S1)$$

where $x_i \in \{0, 1\}$ is the logic of the situation i . $i = 1, 2, \dots, m$. The range of weight for each device cell is $w_{i,j} \in \{-1, 1\}$. $I_{det1} - I_{detn}$ are input to the post-processing module to obtain the determination output. Circuit designs of post-processing modules for analog computing include circuits composed of transimpedance amplifier (TIA), analog-to-digital converter (ADC), digital shifter, and adder²¹, or circuits composed of TIA, analog summation node, and ADC²². Accumulation of determination currents in bit 1– n is conducted in the post-processing module to obtain the optimized value of attention in the given situations:

$$\alpha_{\text{spatial}} = \frac{\sum_{j=1}^n 2^j I_{detj}}{I_{det0}} \quad (S2)$$

where I_{det0} is the normalization constant. α_{spatial} is clamped to the range [0%, 100%].

Detailed discussion of the simulation process is provided in Supplementary Note 5.

Supplementary Figure R13. Modeling of the maximum source current I_{S0} .

Supplementary Figure R14. Flow chart of the attention-enhanced edge intelligence.

References:

21. Jiang, H., Li, W., Huang, S. & Yu, S. A 40nm Analog-Input ADC-Free Compute-in-Memory RRAM Macro with Pulse-Width Modulation between Sub-arrays. in *IEEE Symposium on VLSI Technology and Circuits (VLSI Technology and Circuits)*. 266-267 (IEEE, 2022).
22. Song, W. et al. Programming memristor arrays with arbitrarily high precision for analog computing. *Science* **383**, 903-910 (2024). <https://doi.org/doi:10.1126/science.adi9405>

5. The curve fitting parameters of Fig. 2h and Fig. 2j are missing. Those data may be critical for the modeling of the device.

Response:

We agree with the reviewer that the fitting parameters are critical as a reference for evaluating the simulation process. In the revised version, the fitting parameters are provided in Supplementary Table R3. α_{spatial} and α_{temporal} correspond to the curves in Fig. 2h and Fig. 2j. I_{S0} corresponds to the curve in Supplementary Fig. R13. The device model for edge intelligence simulation is thereby established.

Supplementary Table R3. Exponential fitting parameters.

	Parameters			
	a	b	c	d
I_{S0}	0.130278	-0.0141960	-3.1706	1.20822
α_{spatial}	1.06207	-0.85689	-3.3540	1.26730
α_{temporal}	-0.0036477	1.84917	-5.2985	1.43504

6. The attention is adjusted by V_{CG} in the device in the description of the attention-inspired device principle in Section 3. Is the attention in Section 4 also adjusted by V_{CG} ? If so, the author should give the specific values of the CG voltages corresponding to all the attention values in Fig. 4a, as well as the adjusted attention values in the three scenes in Fig. 4b-d.

Response:

The attention adjustment of the proposed design is by V_{CG} , and the attention adjustment in the edge intelligence application is also based on the tuning of CG voltage. The CG voltages corresponding to attention values in Fig. 4b is shown in Supplementary Table 4. V_{CG} corresponding to attention values in Fig. 4a are shown in Supplementary Table 4. V_{CG} in Fig. 4b-d are shown in Supplementary Table 5. Scene 1 (Wide street) corresponds to Fig. 4b, Scene 2 (Congested street) corresponds to Fig. 4c, and Scene 3 (Crossroad) corresponds to Fig. 4d.

Supplementary Table R4. The spatial attention adjusted by CG voltage.

V_{CG} (V)	-1.33	-1.27	-1.19	-1.08	-0.91	-0.48
$\alpha_{spatial}$	0%	20%	40%	60%	80%	100%

Supplementary Table R6. Spatial attention and CG voltage configurations in different scenes.

	Scene1	Scene2	Scene3
	Wide street	Congested street	Crossroad
$\alpha_{spatial}$	0.5%	47.5%	90.0%
V_{CG} (V)	-1.32	-1.15	-0.77

Figure 4 | Highly adaptive edge equipment and performance projections. **a**, Illustrations of the attention-enhanced infrastructure. Attention can be continuously adjusted from 0 to 100%, and **a** shows examples under 6 values of spatial attention. **b–d**, Illustrations of the attention-enhanced vehicle. Attentions are dynamically adjusted in different scenes, including wide street (**b**), congested street (**c**), and crossroad (**d**). **e**, Latency and **f**, energy projections of the conventional transistor-based circuit and the attention-inspired-device-based architecture for spatial-temporal information processing with attention adjustment from 0% to 100%. **g**, Area, latency, and energy improvements of the attention-inspired device.

Reviewer #3 (Remarks to the Author):

The manuscript J. Pan et al. reports on an attention-inspired artificial intelligence architecture, using the functionalities of Ag/HfO₂/MoS₂ memristive devices and highlight the efficiency compared to conventional transistor based circuits. The manuscript presents interesting results, and the topic is definitely of interest for the journal. However, there are several issues that need to be addressed prior to a decision on the manuscript.

Response:

We sincerely appreciate the reviewer for the positive assessments of our work and constructive suggestions on the initial manuscript. It is very encouraging for us to receive your recognition that “*the topic is definitely of interest for the journal*”. The inconsistencies and comments have been addressed in the revised version. Detailed responses are provided as follows.

Comments:

Whereas I do not have concerns about the computational part, there are several issues explaining the device the authors use.

1. In fact the devices used are typically termed as memristors and more specifically ECM (or CBRAM) memristive devices. This seems to be overlooked in the manuscript.

Response:

The reviewer has pointed out an important issue about the device structure and physical mechanisms. The conductive bridging mechanism is an important part of the proposed device that corresponds to the state transfer of the connection between Ag filament and MoS₂ layer. Besides, there are several characteristics that the proposed device differs from CBRAM memristive devices. First, for CBRAM devices, the top electrode controls the filament-forming and rupture by the applied voltage. For the proposed device, the top electrode controls the filament state and electrostatically tunes the carrier density of the bottom MoS₂ layer simultaneously. The integrated multifunctional mechanism enables the proposed device to

perform dynamic weighted analog computing of both spatial and temporal information. Second, the physical parameters of CBRAM devices (forming/rupture voltages, on/off state resistances, etc.) are typically fixed, in order to ensure stable linear computing in artificial neural networks. On the contrary, features of the proposed device are dynamically adjustable by the buried control gate voltage, thereby enabling dynamic spatial and temporal attention adjustment during time-variant computing. Third, the CBRAM device has two terminals, whereas the proposed device has four terminals (not considering the substrate). The proposed device integrates multidimensional information processing into the device level. The proposed device may not be a typical ECM device, but it is appropriate to categorize the proposed device as a new form of device structure applying the ECM mechanism.

2. The functions and critical parameters of ECM devices are well known, but missing in the manuscript e.g. endurance, retention, switching time, as well the neuromorphics-related characteristics . It should be clarified how reliable are the devices and compare them to state-of-art memristive devices .

Response:

We agree with the reviewer that those functions and parameters are critical for the integrity of this work. We have conducted experiments to measure those parameters. The retention and endurance testing results are shown in Supplementary Fig. 5. The tested retention is larger than 10^4 s. We believe that the actual performance is largely higher than the measured value, since the on-state current degradation of the device at 10^4 s is 1.5%. The tested endurance is larger than 100 cycles.

Supplementary Figure R5. (a) Retention and (b) endurance testing. The read voltage is 0.1 V.

The tested switching time is shown in Supplementary Figure 5b–c. The measured forming time is 190 μ s, and the measured rupture time is 310 μ s. The switching speed is limited by the semiconductor channel. The tested switching time is similar to the performance of the state-of-art memristive device with semiconductor channel (*Nature* **618**, 57-62 (2023)).

Supplementary Figure 4. State transfer time of the attention-inspired device. (a) A cycle of filament forming and rupture. (b) Enlarged waveform of filament forming. (c) Enlarged waveform of filament rupture.

This work is mainly to demonstrate a design to achieve in situ spatial-temporal information processing. To validate our design functionalities, Ag was chosen as the top electrode due to its fabrication viability in our laboratory, and the proposed structure can also be realized by other materials. Although Ag-based resistive switching devices have relatively lower endurance than

VCM including TiN-based devices, several strategies have been reported to enlarge the endurance, including adding a stacking layer structure⁷, widening the intrinsic threading dislocation that acts as the preferential diffusion paths of Ag in the dielectric⁸, reducing the quantity and movement of metal ions during switching via partial filament formation and variations⁹, etc. The performance specifications of the proposed device and state-of-art memristive devices are listed in Supplementary Table R1.

Supplementary Table R1. Comparisons of memristive devices for neuromorphic computing.

References	Endurance	Retention	Switching time	On/Off ratio
Nature 2023 ¹	10 ⁶ cycles	10 ² ~ 10 ⁴ s	232 μ s	10 ²
Nature Materials 2023 ²	10 ⁴ cycles	10 ² s	\	10 ¹⁰
Nature Communications 2019 ³	10 ² cycles	10 ⁴ s	700 ns	10 ³
Nature Communications 2022 ⁴	10 ⁷ cycles	10 ⁵ s	100 ns	10 ²
Nature Communications 2023 ⁵	10 ² cycles	10 ⁴ s	60 ns	10 ²
Nature Communications 2024 ⁶	10 ⁵ cycles	\	ms	2.4
This work	> 10 ² cycles	> 10 ⁴ s	< 310 μ s	10 ⁹

In computing mode, the proposed device behaves as weight for synapse to perform inner production (Fig. R2i). This behavior has been verified to achieve artificial neural networks (*Nature* **577**, 641-646 (2020), *Nat. Nanotechnol.* **17**, 27-32 (2022)).

Figure R2i, Bidirectional determination computing curves of the attention-inspired device.

References:

1. Zhu, K. et al. Hybrid 2D-CMOS microchips for memristive applications. *Nature* **618**, 57-62 (2023). <https://doi.org/10.1038/s41586-023-05973-1>
2. Kang, J.-H. et al. Monolithic 3D integration of 2D materials-based electronics towards ultimate edge computing solutions. *Nature Materials* **22**, 1470-1477 (2023). <https://doi.org/10.1038/s41563-023-01704-z>
3. Sivan, M. et al. All WSe₂ 1T1R resistive RAM cell for future monolithic 3D embedded memory integration. *Nat. Commun.* **10**, 5201 (2019). <https://doi.org/10.1038/s41467-019-13176-4>
4. Tang, B. et al. Wafer-scale solution-processed 2D material analog resistive memory array for memory-based computing. *Nat. Commun.* **13**, 3037 (2022). <https://doi.org/10.1038/s41467-022-30519-w>
5. Xie, M. et al. Monolithic 3D integration of 2D transistors and vertical RRAMs in 1T-4R structure for high-density memory. *Nat. Commun.* **14**, 5952 (2023). <https://doi.org/10.1038/s41467-023-41736-2>
6. Park, J. et al. Multi-level, forming and filament free, bulk switching trilayer RRAM for neuromorphic computing at the edge. *Nat. Commun.* **15** (2024). <https://doi.org/10.1038/s41467-024-46682-1>
7. Yan, X. et al. Robust Ag/ZrO₂/WS₂/Pt Memristor for Neuromorphic Computing. *ACS Appl Mater Interfaces* **11**, 48029-48038 (2019). <https://doi.org/10.1021/acsami.9b17160>

8. Choi, S. et al. SiGe epitaxial memory for neuromorphic computing with reproducible high performance based on engineered dislocations. *Nat. Mater.* **17**, 335-340 (2018). <https://doi.org/10.1038/s41563-017-0001-5>
9. Kim, J., Kwon, O., Seo, J. & Hwang, H. Vertical - Switching Conductive Bridge Random Access Memory with Adjustable Tunnel Gap and Improved Switching Uniformity Using 2D Electron Gas. *Advanced Electronic Materials* (2024). <https://doi.org/10.1002/aelm.202400650>

3. It is also well-known that oxide-type ECM memristors are sensitive to water molecules, protons/moisture coming from surroundings and/or during device preparation steps. Moisture effects can significantly alter the device performance. Could the authors comment on this?

Response:

The reviewer has raised a critical discussion that should be included in this work. Moisture effect is one of the main effects that influence ECM memristors performance. H₂O molecules are incorporated by absorption within the oxide with or without chemical interaction/dissociation, or defect-chemical reaction that introduces protons within the oxide¹². Moisture has multiple effects on the dielectric film and device properties. The presence of H₂O molecules enables the filament formation, and affects the forming voltage¹³⁻¹⁶. Moisture also affects the switching behavior and set/reset kinetics¹⁷⁻¹⁹. It has been reported that moisture effects are relevant to matrix film properties including density and composition^{20,21}. Furthermore, the retention performance is reduced with the increase of humidity²². To enhance the devices' stability, the effects of moisture should be controlled. Discussions on moisture effects are added in Supplementary Note 2 and Lines 161–163 of the main text:

The performance of device can also be affected by external factors including proton or moisture incorporated into the dielectric³⁷⁻³⁹. More discussions on device functionality stability are provided in Supplementary Note 2.

Reference (Supplementary Information):

10. Valov, I. & Tsuruoka, T. Effects of moisture and redox reactions in VCM and ECM resistive switching memories. *Journal of Physics D: Applied Physics* **51** (2018). <https://doi.org/10.1088/1361-6463/aad581>
11. Tsuruoka, T. et al. Effects of Moisture on the Switching Characteristics of Oxide-Based, Gapless - Type Atomic Switches. *Adv. Funct. Mater.* **22**, 70-77 (2011). <https://doi.org/10.1002/adfm.201101846>
12. Tappertzhofen, S., Hempel, M., Valov, I. & Waser, R. Proton mobility in SiO₂ thin films and impact of hydrogen and humidity on the resistive switching effect. *MRS Proceedings* **1330** (2011). <https://doi.org/10.1557/opl.2011.1198>
13. Tappertzhofen, S. et al. Generic relevance of counter charges for cation-based nanoscale resistive switching memories. *ACS Nano* **7**, 6396-6402 (2013). <https://doi.org/10.1021/nn4026614>
14. Tsuruoka, T., Hasegawa, T., Terabe, K. & Aono, M. Operating mechanism and resistive switching characteristics of two- and three-terminal atomic switches using a thin metal oxide layer. *Journal of Electroceramics* **39**, 143-156 (2017). <https://doi.org/10.1007/s10832-016-0063-9>
15. Ngaruiya, J. M., Kappertz, O., Mohamed, S. H. & Wuttig, M. Structure formation upon reactive direct current magnetron sputtering of transition metal oxide films. *Applied Physics Letters* **85**, 748-750 (2004). <https://doi.org/10.1063/1.1777412>
16. Chang, C.-F. et al. Direct Observation of Dual-Filament Switching Behaviors in Ta₂O₅-Based Memristors. *Small* **13**, 1603116 (2017). <https://doi.org/https://doi.org/10.1002/smll.201603116>
17. Lubben, M. et al. SET kinetics of electrochemical metallization cells: influence of counter-electrodes in SiO₂/Ag based systems. *Nanotechnology* **28**, 135205 (2017). <https://doi.org/10.1088/1361-6528/aa5e59>
18. Tsuruoka, T. et al. Redox Reactions at Cu,Ag/Ta₂O₅ Interfaces and the Effects of Ta₂O₅ Film Density on the Forming Process in Atomic Switch Structures. *Adv. Funct. Mater.* **25**, 6374-6381 (2015). <https://doi.org/https://doi.org/10.1002/adfm.201500853>
19. Mannequin, C., Tsuruoka, T., Hasegawa, T. & Aono, M. Identification and roles of nonstoichiometric oxygen in amorphous Ta₂O₅ thin films deposited by electron beam

and sputtering processes. *Applied Surface Science* **385**, 426-435 (2016).
<https://doi.org/10.1016/j.apsusc.2016.04.099>

Reference (Main Text):

37. Valov, I. & Tsuruoka, T. Effects of moisture and redox reactions in VCM and ECM resistive switching memories. *Journal of Physics D: Applied Physics* **51** (2018).
<https://doi.org/10.1088/1361-6463/aad581>
38. Tsuruoka, T. et al. Effects of Moisture on the Switching Characteristics of Oxide-Based, Gapless - Type Atomic Switches. *Adv. Funct. Mater.* **22**, 70-77 (2011).
<https://doi.org/10.1002/adfm.201101846>
39. Ngaruiya, J. M., Kappertz, O., Mohamed, S. H. & Wuttig, M. Structure formation upon reactive direct current magnetron sputtering of transition metal oxide films. *Applied Physics Letters* **85**, 748-750 (2004). <https://doi.org/10.1063/1.1777412>

4. It should be discussed and clarified, what is the reason (and advantages) on taking the Ag/HfO₂/MoS₂ system. What is the role of the materials composition and thicknesses. For example it should be explained what Ag is selected and not (for example Cu), as well why HfO₂ and not e.g. TaO_x or SiO₂, etc.

Response:

Ag⁺ ions have larger diffusivity in oxide films than inert metals such as Ni and Pt, which means that the filament-forming and rupture processes are easier to achieve (*Nat. Commun.* **5**, 4232 (2014)). Ag has a large conductivity, so the on-state resistance is low and the on/off state ratio is large. Furthermore, Ag filament has been reported to have stable contact with 2D transition metal dichalcogenides (TMD) including MoS₂. The Ag filament connection between and 2D TMD layer can be repeatedly formed and ruptured with high stability, and exhibits good contact properties (*ACS Nano* **13**, 2205-2212 (2019)). To our knowledge, there is no report on the contact between Cu filament and 2D TMD layers. Therefore, Ag is compatible with the proposed device design. HfO₂ is used in this work because it is a high-k dielectric with a large

dielectric constant (*J. Vac. Sci. Technol. B* **18**, 1785-1791 (2000)), which enables the top Ag electrode to perform active electrostatically gating to the bottom MoS₂ channel within relative low voltage levels. Moreover, the ALD process of HfO₂ does not affect the performance of the MoS₂ layer (*Nanotechnology* **29**, 345201 (2018)). Therefore, HfO₂ was chosen as the top dielectric of the MoS₂ layer.

5. How the selected devices facilitate the observed advanced functionalities.

Response:

The proposed device has multiple states by the forming and rupture processes of the interfacial connections between MoS₂ and Ag filament, enabling non-volatile storage of information (Fig. 2b), and the on-state characteristics can be modulated by the buried control gate voltage (Fig. 2d). For different filament states, the proposed device exhibits diverse transport characteristics that are different from conventional transistors (Fig. 2f and Fig. 2i), which expand the limitation of information processing capacity, and can be used to facilitate spatial-temporal information perception. For attention distribution computing, the transport behavior in Fig. 2f can be applied to establish weighted mappings between the input, stored, and output data. The mapping characteristics can be continuously adjusted by the CG voltage. With the increase of the CG voltage, spatial information is increased, and temporal information is decreased (Fig. 2g). Spatial and temporal attention are thereby defined to describe the property, which have opposite variation trends with CG voltage (Fig. 2h). Therefore, Intensities of spatial and temporal information can be continuously adjusted by the control terminal. For attention distribution computing, the transport behavior in Fig. 2i can be applied to define bipolar weight values by the filament states, which is reconfigurable (Fig. 2j), and can be used for the inner production and matrix multiplication of decision-making computing.

Figure 2 | Working mechanisms of the attention-inspired device. Writing mode: **a**, Output characteristics of the MoS₂ channel. $V_{DS} = 1.00$ V. **b**, Attention-inspired device filament state transfer curve. The voltage scanning path is from 1 to 4. $V_{CG} = 5.00$ V. **c**, Writing mode configuration. **d**, Filament shunt current modulation characteristics with cut-off, gate-controlled, and resistive regions varying with V_{CG} . **e**, Structure of an attention-inspired device working in writing and computing modes. Computing mode (attention distribution computing): **f**, I_S - V_{IN} transport curves under different V_{CG} from -1.80 to 0.00 V. Device structure schematics illustrate states of the attention-inspired device and current directions under each voltage configuration. IN and CG electrode voltage configurations are represented by blue (low voltage), red (high voltage), or gray (arbitrary voltage) colors. **g**, I_S - V_{CG} attention distribution characteristics. $V_{IN-} = -0.72$ V, $V_{IN+} = -0.36$ V. **h**, Spatial and temporal attention varying with V_{CG} that exhibit dynamic attention adjustment properties. Dash lines show the exponential fitting. $V_S = -0.20$ V in f-h. Computing mode (determination computing): **i**, Bidirectional determination computing curves of the attention-inspired device. **j**, Weight plasticity adjusted by V_{IN+} . The dash line shows the linear fitting. $V_S = 0.20$ V and $V_{CG} = 0.60$ V in i-j. $V_D = 0.00$ V and $V_B = -1.00$ V in f-j.

6. The operation mechanism needs more clear explanation and justification.

Response:

The proposed device induces 2D electrostatic modulation interfaces between MoS₂, HfO₂ and Ag layers, 0D contact interfaces between MoS₂ channel and Ag filament. The top Ag electrode controls the filament state and electrostatically modulates the carrier density of the bottom MoS₂ layer simultaneously. The integrated hetero-dimensional modulation mechanism extends the control dimension at the device level, and the hetero-dimensional interfaces' characteristics can be continuously adjusted by the buried control gate, which enables the device to have dynamic adaptability. When performing the operations, the output current is modulated by the absolute voltages at both the top Ag electrode and control gate terminal, the potential barrier formed near the filament contact interface, and the shunt current at the Ag electrode. Multiple effects are thereby induced, including the saturation characteristics (exclusive control ability of the Ag electrode) in w/o-filament state, bidirectional current modulations, and current approximation effect, which are combined to form the transport characteristics of the proposed device (Fig. 2f and Fig. 2i). Therefore, spatial-temporal mapping between the input voltage and output current is built, and continuously adjusted by a control gate, and in situ weighted analog computing can be performed in the device, which leads to the complete and adaptive spatial-temporal information perception. The main text has been revised for more clear explanation and justification:

Lines 64–67:

Zero-dimensional (0D) interface exhibits non-volatile state transfer behavior for data storage, and adjustable weighted analog computing is performed between input and stored data based on intrinsic interactions of 0D-2D hetero-dimensional interfaces.

Lines 164–169:

In computing mode, the attention-inspired device exhibits reconfigurable properties to implement attention distribution and determination computing modes by the source voltage V_S

and the CG voltage V_{CG} . V_{CG} is tunable to adjust the attention. For attention distribution computing, transport curves varying with V_{CG} in different filament states to emulate attention-directed perception in the brain are illustrated in Fig. 2f (positive direction of I_S is from drain to source).

Lines 174–177:

When $V_{CG} > V_{IN}$, the homojunction barrier is forwardly biased. With fixed V_{IN} , the electron carrier density is unchanged by V_{CG} , and I_S is saturated. The saturation characteristics in w/o-filament state provide an interval of V_{CG} in which I_S is slightly varied.

Lines 182–189:

As V_{IN} is reduced to -0.8 V, ϕ_{0D} is decreased, and the potential difference between 0D interface and source is reversed by negative ϕ_{0D} , so negative I_S is exhibited (light red lines in Fig. 2f). When V_{IN} is increased, the shunt current is reduced and the difference between ϕ_{0D} and V_S is decreased, so the absolute of negative I_S is reduced and eventually reach to zero. With the increase of V_{CG} , electron carrier flow from 0D interface to drain is enhanced by lowered band of the CG-controlled channel, ϕ_{0D} approaches to or exceeds V_S , and the I_S – V_{IN} transport curve is lifted (from light to dark red lines in Fig. 2f). For $V_{CG} > -0.60$ V, ϕ_{0D} is close to V_D , so the curve of I_S approximates to the transfer curve of w/o-filament state.

Lines 203–213:

Bidirectional I_S responses enable bipolar weighted analog computing (Fig. 2i). In w/o-filament state, 2D homojunction is formed by the gating of V_{IN} . For negative V_{IN} ($V_{IN} < -0.50$ V), the homojunction potential barrier is reversely biased to cut off the channel. For positive V_{IN} ($V_{IN} > 0.20$ V), the barrier is forwardly biased, and on-state I_S is observed (the blue line in Fig. 2i). In w/-filament state, when $V_{IN} < -0.50$ V, negative shunt current is imported from 0D interface, and ϕ_{0D} is reduced. A large V_{CG} (0.6 V) bends down the CG-controlled channel and increases electron carrier density from 0D interface to drain, which attenuates the potential reduction effect of ϕ_{0D} induced by shunt current. For $V_{IN} < -0.50$ V, the homojunction potential barrier is reversely biased, and I_S is reduced to the sub-threshold level. When $V_{IN} > 1.30$ V, positive shunt

currents are induced, ϕ_{0D} is larger than V_s , and positive I_s from 0D interface to source is observed (the red line in Fig. 2i).

The above mentioned points should be implemented in the text supported by corresponding citations from the literature.

Response:

Thank you for your constructive comments that help to enhance the quality of this work. We have addressed the issues, and have added 37 articles as supporting evidence during the discussion, in order to enhance the integrity of discussions and analyses. The added references are as follows:

Added References in the main text:

2. Serrano-Gotarredona, T. & Linares-Barranco, B. A 128×128 1.5% Contrast Sensitivity 0.9% FPN 3 μ s Latency 4 mW Asynchronous Frame-Free Dynamic Vision Sensor Using Transimpedance Preamplifiers. *IEEE J. Solid-State Circuits* **48**, 827-838 (2013). <https://doi.org/10.1109/jssc.2012.2230553>
3. Choo, K. D. et al. Energy-Efficient Low-Noise CMOS Image Sensor with Capacitor Array-Assisted Charge-Injection SAR ADC for Motion-Triggered Low-Power IoT Applications. in *Proc. IEEE International Solid-State Circuits Conference - (ISSCC)*. 96-98 (IEEE, 2019).
14. Huang, X. et al. An ultrafast bipolar flash memory for self-activated in-memory computing. *Nat. Nanotechnol.* **18**, 486-492 (2023). <https://doi.org/10.1038/s41565-023-01339-w>
23. Tan, H. & van Dijken, S. Dynamic machine vision with retinomorphic photomemristor-reservoir computing. *Nat. Commun.* **14**, 2169 (2023). <https://doi.org/10.1038/s41467-023-37886-y>

25. Huang, H. et al. In-sensor compressing via programmable optoelectronic sensors based on van der Waals heterostructures for intelligent machine vision. *Nat. Commun.* **16**, 3836 (2025). <https://doi.org/10.1038/s41467-025-59104-7>
28. Zhu, X. et al. High-Contrast Bidirectional Optoelectronic Synapses based on 2D Molecular Crystal Heterojunctions for Motion Detection. *Adv. Mater.* **35**, e2301468 (2023). <https://doi.org/10.1002/adma.202301468>
29. Dang, Z. et al. Object Motion Detection Enabled by Reconfigurable Neuromorphic Vision Sensor under Ferroelectric Modulation. *ACS Nano* **18**, 27727-27737 (2024). <https://doi.org/10.1021/acsnano.4c10231>
32. Park, E. et al. Quasi-Zero-Dimensional Source/Drain Contact for Fermi-Level Unpinning in a Tungsten Diselenide (WSe₂) Transistor: Approaching Schottky-Mott Limit. *ACS Nano* **18**, 29771-29778 (2024). <https://doi.org/10.1021/acsnano.4c09384>
34. Yan, X. et al. Robust Ag/ZrO₂/WS₂/Pt Memristor for Neuromorphic Computing. *ACS Appl. Mater. Interfaces* **11**, 48029-48038 (2019). <https://doi.org/10.1021/acscami.9b17160>
35. Choi, S. et al. SiGe epitaxial memory for neuromorphic computing with reproducible high performance based on engineered dislocations. *Nat. Mater.* **17**, 335-340 (2018). <https://doi.org/10.1038/s41563-017-0001-5>
36. Kim, J., Kwon, O., Seo, J. & Hwang, H. Vertical - Switching Conductive Bridge Random Access Memory with Adjustable Tunnel Gap and Improved Switching Uniformity Using 2D Electron Gas. *Adv. Electron. Mater.* (2024). <https://doi.org/10.1002/aelm.202400650>
37. Valov, I. & Tsuruoka, T. Effects of moisture and redox reactions in VCM and ECM resistive switching memories. *J. Phys. D: Appl. Phys.* **51** (2018). <https://doi.org/10.1088/1361-6463/aad581>
38. Tsuruoka, T. et al. Effects of Moisture on the Switching Characteristics of Oxide - Based, Gapless - Type Atomic Switches. *Adv. Funct. Mater.* **22**, 70-77 (2011). <https://doi.org/10.1002/adfm.201101846>
39. Ngaruiya, J. M., Kappertz, O., Mohamed, S. H. & Wuttig, M. Structure formation upon reactive direct current magnetron sputtering of transition metal oxide films. *Appl. Phys. Lett.* **85**, 748-750 (2004). <https://doi.org/10.1063/1.1777412>

Added References in Supplementary Information:

1. Zhu, K. et al. Hybrid 2D-CMOS microchips for memristive applications. *Nature* **618**, 57-62 (2023). <https://doi.org/10.1038/s41586-023-05973-1>
2. Kang, J.-H. et al. Monolithic 3D integration of 2D materials-based electronics towards ultimate edge computing solutions. *Nat. Mater.* **22**, 1470-1477 (2023). <https://doi.org/10.1038/s41563-023-01704-z>
3. Sivan, M. et al. All WSe₂ 1T1R resistive RAM cell for future monolithic 3D embedded memory integration. *Nat. Commun.* **10**, 5201 (2019). <https://doi.org/10.1038/s41467-019-13176-4>
4. Tang, B. et al. Wafer-scale solution-processed 2D material analog resistive memory array for memory-based computing. *Nat. Commun.* **13**, 3037 (2022). <https://doi.org/10.1038/s41467-022-30519-w>
5. Xie, M. et al. Monolithic 3D integration of 2D transistors and vertical RRAMs in 1T-4R structure for high-density memory. *Nat. Commun.* **14**, 5952 (2023). <https://doi.org/10.1038/s41467-023-41736-2>
6. Park, J. et al. Multi-level, forming and filament free, bulk switching trilayer RRAM for neuromorphic computing at the edge. *Nat. Commun.* **15** (2024). <https://doi.org/10.1038/s41467-024-46682-1>
7. Yan, X. et al. Robust Ag/ZrO₂/WS₂/Pt Memristor for Neuromorphic Computing. *ACS Appl. Mater. Interfaces* **11**, 48029-48038 (2019). <https://doi.org/10.1021/acsami.9b17160>
8. Choi, S. et al. SiGe epitaxial memory for neuromorphic computing with reproducible high performance based on engineered dislocations. *Nat. Mater.* **17**, 335-340 (2018). <https://doi.org/10.1038/s41563-017-0001-5>
9. Kim, J., Kwon, O., Seo, J. & Hwang, H. Vertical - Switching Conductive Bridge Random Access Memory with Adjustable Tunnel Gap and Improved Switching Uniformity Using 2D Electron Gas. *Adv. Electron. Mater.* (2024). <https://doi.org/10.1002/aelm.202400650>
10. Valov, I. & Tsuruoka, T. Effects of moisture and redox reactions in VCM and ECM resistive switching memories. *J. Phys. D: Appl. Phys.* **51** (2018). <https://doi.org/10.1088/1361-6463/aad581>

11. Tsuruoka, T. et al. Effects of Moisture on the Switching Characteristics of Oxide - Based, Gapless - Type Atomic Switches. *Adv. Funct. Mater.* **22**, 70-77 (2011). <https://doi.org/10.1002/adfm.201101846>
12. Tappertzhofen, S., Hempel, M., Valov, I. & Waser, R. Proton mobility in SiO₂ thin films and impact of hydrogen and humidity on the resistive switching effect. *Mater. Res. Soc. Symp. Proc.* **1330** (2011). <https://doi.org/https://doi.org/10.1557/opl.2011.1198>
13. Tappertzhofen, S. et al. Generic relevance of counter charges for cation-based nanoscale resistive switching memories. *ACS Nano* **7**, 6396-6402 (2013). <https://doi.org/10.1021/nn4026614>
14. Tsuruoka, T., Hasegawa, T., Terabe, K. & Aono, M. Operating mechanism and resistive switching characteristics of two- and three-terminal atomic switches using a thin metal oxide layer. *J. Electroceramics* **39**, 143-156 (2017). <https://doi.org/10.1007/s10832-016-0063-9>
15. Ngaruiya, J. M., Kappertz, O., Mohamed, S. H. & Wuttig, M. Structure formation upon reactive direct current magnetron sputtering of transition metal oxide films. *Appl. Phys. Lett.* **85**, 748-750 (2004). <https://doi.org/10.1063/1.1777412>
16. Chang, C.-F. et al. Direct Observation of Dual-Filament Switching Behaviors in Ta₂O₅-Based Memristors. *Small* **13**, 1603116 (2017). <https://doi.org/https://doi.org/10.1002/sml.201603116>
17. Lubben, M. et al. SET kinetics of electrochemical metallization cells: influence of counter-electrodes in SiO₂/Ag based systems. *Nanotechnology* **28**, 135205 (2017). <https://doi.org/10.1088/1361-6528/aa5e59>
18. Tsuruoka, T. et al. Redox Reactions at Cu,Ag/Ta₂O₅ Interfaces and the Effects of Ta₂O₅ Film Density on the Forming Process in Atomic Switch Structures. *Adv. Funct. Mater.* **25**, 6374-6381 (2015). <https://doi.org/https://doi.org/10.1002/adfm.201500853>
19. Mannequin, C., Tsuruoka, T., Hasegawa, T. & Aono, M. Identification and roles of nonstoichiometric oxygen in amorphous Ta₂O₅ thin films deposited by electron beam and sputtering processes. *Appl. Surf. Sci.* **385**, 426-435 (2016). <https://doi.org/10.1016/j.apsusc.2016.04.099>

20. Mannequin, C., Tsuruoka, T., Hasegawa, T. & Aono, M. Composition of thin Ta₂O₅ films deposited by different methods and the effect of humidity on their resistive switching behavior. *Jpn. J. Appl. Phys.* **55** (2016). <https://doi.org/10.7567/jjap.55.06gg08>
21. Jiang, H., Li, W., Huang, S. & Yu, S. A 40nm Analog-Input ADC-Free Compute-in-Memory RRAM Macro with Pulse-Width Modulation between Sub-arrays. in *Proc. IEEE Symposium on VLSI Technology and Circuits (VLSI Technology and Circuits)*. 266-267 (IEEE, 2022).
22. Song, W. et al. Programming memristor arrays with arbitrarily high precision for analog computing. *Science* **383**, 903-910 (2024). <https://doi.org/doi:10.1126/science.adi9405>
28. Rabaey, J. M., Chandrakasan, A. & Nikolić, B. *Digital integrated circuits: a design perspective* Ed. 2 (Pearson, 2002).